# GraphTS: Graph-represented time series for subsequence anomaly detection

**Roozbeh Zarei, Guangyan Huang\*, Junfeng Wu**

School of Information Technology, Deakin University, Melbourne, Victoria, Australia

\* guangyan.huang@deakin.edu.au

## Abstract

Automatic detection of subsequence anomalies (i.e., an abnormal waveform denoted by a sequence of data points) in time series is critical in a wide variety of domains. However, most existing methods for subsequence anomaly detection often require knowing the length and the total number of anomalies in time series. Some methods fail to capture recurrent subsequence anomalies due to using only local or neighborhood information for anomaly detection. To address these limitations, in this paper, we propose a novel graph-represented time series (GraphTS) method for discovering subsequence anomalies. In GraphTS, we provide a new concept of time series graph representation model, which represents both recurrent and rare patterns in a time series. Particularly, in GraphTS, we develop a new 2D time series visualization (2Dviz) method, which compacts all 1D time series patterns into a 2D spatial temporal space. The 2Dviz method transfers time series patterns into a higher-resolution plot for easier sequence anomaly recognition (or detecting subsequence anomalies). Then, a Graph is constructed based on the 2D spatial temporal space of time series to capture recurrent and rare subsequence patterns effectively. The represented Graph also can be used to discover single and recurrent subsequence anomalies with arbitrary lengths. Experimental results demonstrate that the proposed method outperforms the state-of-the-art methods in terms of accuracy and efficiency.

## 1 Introduction

Time series anomaly detection is an important problem with applications in various domains such as manufacturing, medical, and engineering [1–9]. Generally, an anomaly changing with time [10] can be a point anomaly (i.e., a single value beyond a regular range) or a sequence anomaly (i.e., an abnormal waveform denoted by a sequence of data points) [11, 12]. Detecting sequence anomalies is crucial, especially in real-world applications, where the values of individual points may not exhibit any anomaly but the trend (or the shape of a subsequence) may be abnormal. More significantly, an abnormal trend often means a possible problem at an early stage that may lead to a severe problem if not intervened. For instance, detecting some abnormal heartbeat waveforms (i.e., arrhythmia) in electrocardiograms may indicate an early stage of a severe heart disease. Therefore, this paper focuses on detecting subsequence anomalies.

**Data Availability Statement:** The data sets used in the evaluation section are publicly available: the MIT-BIH Supraventricular Arrhythmia Database (svdb) and Arrhythmia Database (mitdb) are available from the following repositories (https://physionet.org/content/svdb/1.0.0/, https://

physionet.org/content/mitdb/1.0.0/). The simulated engine disks dataset (SED) is available from https://data.nasa.gov/dataset/Rotor-health-monitoring-combining-spin-tests-and-d/rbn3-kay3.

**Funding:** This work was partially supported by Australia Research Council (ARC) Discovery Project (DP190100587, https://www.arc.gov.au/, GH). The funders had no role in study design, data collection and analysis, decision to publish, or preparation of the manuscript.

**Competing interests:** The authors have declared that no competing interests exist.

Unfortunately, automatically detecting subsequence anomalies faces three challenges. First, most existing subsequence anomaly detection methods only work in a specific domain-determined waveform, such as signals of electrocardiogram (ECG) [13] or electroencephalogram (EEG) [14, 15]. They involve specific domain knowledge about the waveform and length of the anomaly to discover anomalous subsequences. As the characteristics of the anomalies (i.e., waveform patterns and lengths) in different domains are often different, it is hard to apply these domain-specific techniques to another domain. Second, several domain-agnostic methods [4, 16] specifically developed for detecting subsequence anomalies in diverse domains demonstrate an inability to identify repeat anomalies comprising highly similar instances of anomalous subsequences [17–19]. These methods use local or neighborhood information to define subsequence anomalies. For example, they often use the largest distances of subsequence to its nearest neighbors to identify anomalies. The assumption behind these methods is that the abnormal subsequence is distant (i.e., entirely separated) from the normal subsequences; that is, if a subsequence pattern has at least two instances, it is not abnormal. Therefore, these methods can detect a single abnormal subsequence or multiple dissimilar abnormal subsequences (referred to as discords) in time series. However, they fail to detect those recurrent abnormal subsequences (called the "twin freak" problem [19]) with similar shapes. To solve this issue, the $m^{th}$ nearest neighbor can be used instead of the first nearest neighbor to calculate the discord score. In [20], an abnormal subsequence pattern with $m$ instances can be identified. But, this method implies that the number of anomalies is known; this is hard to be satisfied in reality. Third, existing methods for discord discovery can accurately find anomalies when the proper subsequence length is selected [21] as input parameter by the user but suffers noticeably when the length is mismatched. To show how these methods are impacted by subsequence length, we consider two subsequence lengths, 50 and 70, to compute anomaly score using STOMP [21] for every subsequence on a time series as shown in Fig 1(a). The time series is the Arterial Blood Pressure (ABP) of a healthy man on title table with one synthetic anomaly (highlighted in red area in Fig 1(a)). For a subsequence length of 70 as shown in Fig 1(b), the anomaly is correctly identified as indicated by the highest discord score. However, for subsequence length of 50 as shown in Fig 1(c), the normal part of the signal is identified as an anomaly, therefore it is a false positive.

To address the aforementioned three challenges, we propose a novel graph-represented time series (GraphTS) method for subsequence anomaly detection. The GraphTS represents time series as a graph that is constructed using normal and abnormal time series subsequences. In GraphTS, we first develop a new 2D visualization (2Dviz) method, which transfers a time series into a 2-dimensional spatial-temporal space (2DSTS) by projecting subsequences with similar patterns into very close spatial locations. Then, the spatial and temporal information in 2DSTS is used to construct a graph, where nodes represent subsequence patterns and edges represent the number of successive occurrences of these patterns in the original time series. The constructed graph represents all subsequences in time series, including regular, frequent, and anomalous patterns. The recurrent consecutive normal patterns and rare abnormal patterns in time series are represented by paths in the graph that are composed of high- and low-weighted edges, respectively. This enables distinguishing between normal subsequences from abnormal ones using the represented graph.

One advantage of GraphTS is that the graph representation of time series captures both subsequence patterns (i.e., recurrent patterns and abnormal patterns) and thus can detect a complete set of subsequence anomalies. Another advantage of our GraphTS method is its ability to simplify the task of detecting anomalous subsequences by converting raw time series data into a graph representation; so, anomalous subsequences are those with low weights on the path's edges between two nodes. The third advantage is that GraphTS builds the graph

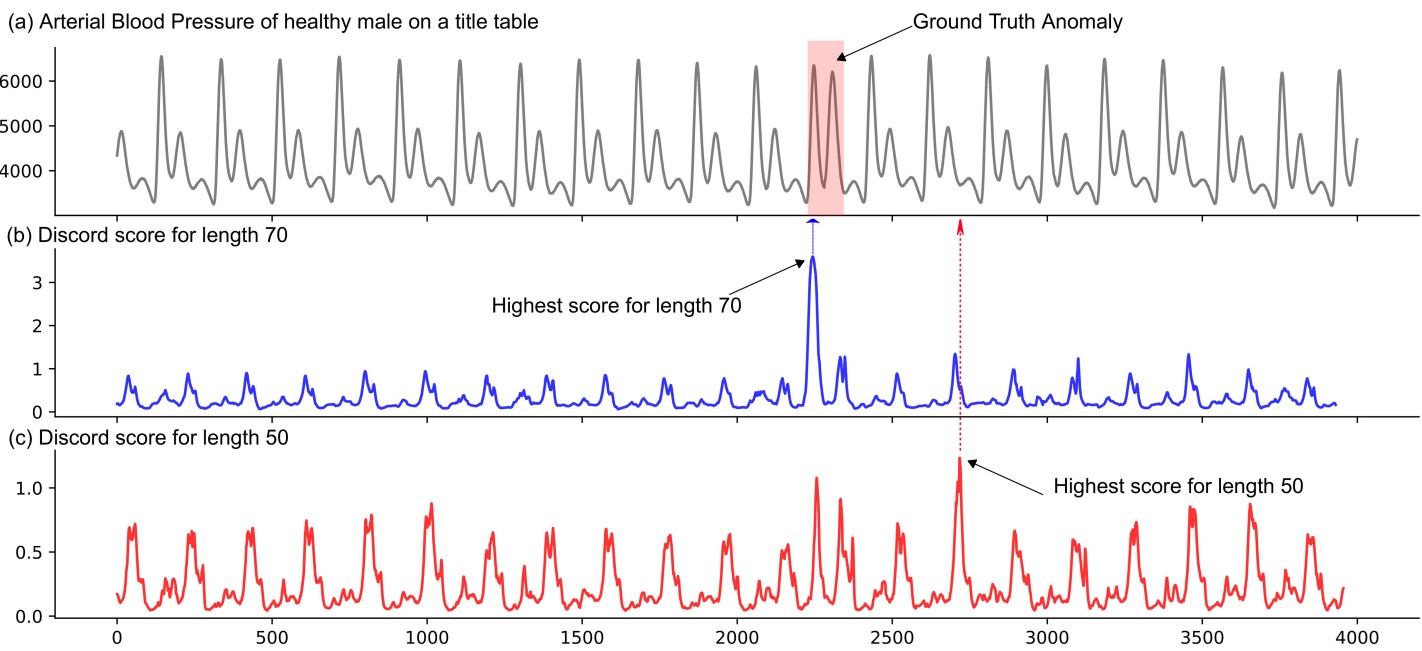

**Fig 1. Importance of subsequence length in anomaly detection.** (a) the ABP time series of a healthy man on a tilt table test with one anomaly highlighted in red. The discord score for each subsequence of length 70 (b) and 50 (c). The corresponding subsequence with the highest score is considered an anomaly.

without knowing the length of the anomalies and it can identify anomalous sequences with arbitrary lengths using the same representation graph. Experimental results show that the proposed method outperforms the state-of-the-art STOMP [21] and Series2Graph [22] methods in terms of both accuracy and execution time.

Our proposed method is different from the two most related works (STOMP and Series2-Graph), and we explain as follows. (1) STOMP detects anomalies by defining local discord patterns; therefore, it has the limitation of being unable to detect recurrent anomalies. Our proposed GraphTS globally project similar subsequences on time series into close nodes in a 2D graph and thus can correctly detects both single and frequent anomalies based on its representation path on the graph. (2) STOMP needs to execute for each anomaly length $\ell$, and its performance is degraded if the value of $\ell$ is not correctly selected. Our method compacts all global information of a time series into a graph that allows identifying anomalies with different lengths, and thus, it is robust to variation of anomaly length. (3) While Series2Graph also utilizes a graph representation to identify anomalies, it adopts an entirely different approach to create the graph, which will be elaborated in Section 2. Our method utilizes the length of normal patterns to construct a graph, ensuring that the variation in anomaly length $\ell$ does not affect its performance.

Our contributions in this paper can be summarized as follows.

1. We develop a novel 2D time series visualization (2Dviz) method, which can compact all patterns on one-dimensional time series into a 2D spatial-temporal space, where time series patterns and anomalies are mapped in a higher-resolution 2Dviz plot for much easier subsequence anomaly recognition.

2. We propose a novel GraphTS method for domain agnostic subsequence anomaly detection. In GraphTS, we provide a new time series graph representation model, which represents

both recurrent and rare patterns in time series. GraphTS can effectively detect recurrent and single subsequence anomalies in an unsupervised way (without knowing the length and the total number of anomalies) and can be used to discover variable-length subsequence anomalies.

3. We demonstrate the accuracy and efficiency of GraphTS by comparing to two state-of-the-art methods (i.e., STOMP and Series2Graph) on real-world time series datasets containing single and multiple recurrent subsequence anomalies.

The remainder of this paper is organized as follows. Section 2 presents the related work. In section 3, we define the problem. We detail the GraphTS method for subsequence anomaly detection using graph representation in section 4. Section 5 reports the experimental results over various real datasets, and section 6 concludes this paper.

## 2 Related work

As our focus is to detect anomalous subsequence in time series based on graph representation in this paper, we present a brief review of subsequence anomaly detection methods and time series graph representation for anomaly detection.

**A. Subsequence anomaly detection.** The problem of detecting subsequence anomalies from time series has been studied in several works based on the discord definition [4, 23–28]. In these methods, anomalous subsequences (discord) are identified based on their distances to all other subsequences in the time series. Specifically, the subsequence with the largest Euclidean distance to its nearest neighbors is considered as a discord or anomaly. These discord discovery algorithms can be applied either on original raw values [4, 25, 28] or on a representation of the subsequences such as Symbolic aggregate approximation (SAX) [23, 24] or Haar wavelets [26, 27].

HOTSAX is developed in [29] based on the SAX representation to detect the time series discord. The SAX is a method for converting time series data into a symbolic representation. The SAX symbolizes the subsequence by the mean of each subsequence's segment; however, due to dimension reduction, it may omit crucial patterns in the subsequence. Several algorithms have been developed to improve the SAX representation of time series [30, 31]. Extended SAX (ESAX) is developed based on SAX in [30] by adding two extra points, min and max points, to each subsequence segment's mean value for improving SAX representation. Trend Distance (SAX-TD) [31] integrates the SAX distance with a weighted trend distance to improve SAX representation. It computes the distance of trends using segments' starting and ending points. Although both ESAX and SAX-TD methods improve the original SAX representation, they still may lose important time series pattern information due to dimension reduction. Senin et al. [16, 23] developed GrammarViz method based on grammar compression of time series discretized with SAX to detect time series discord. Subsequences correspond to rare grammar rules are considered discord as their SAX symbols are not compressible and most likely to be rare patterns. The performance of these discord discovery algorithms based on SAX is heavily reliant on the quality of the SAX data representation. The SAX method has three user-defined parameters that need to carefully set in order to get proper representation. However, selecting and fine-tuning these parameters are generally not trivial.

Recently, a matrix-profile-based approach, STOMP [21], is developed that computes the matrix profile that allows discovering top-1 discord. The STOMP provides a fast calculation of the distance of each subsequence to its nearest non-self neighbor. The main advantage of the above methods is their simplicity. However, these methods are not able to detect repeated anomalies with the same shape in different instances. To resolve the problem of detecting

recurrent (repeated) anomalies by discovering discord, the notion of $m^{th}$ discord is proposed in [20], known as DRAG, in which the subsequence is a discord if it has the largest Euclidean distances to its $m^{th}$ nearest neighbor. The DRAG algorithm is separated into two steps. The initial step involves selecting potential discord sequences by identifying those whose distances to their nearest neighbors fall within a certain range. The second step, known as refinement, is then used to determine the precise discord sequences from among the candidates identified in the first step. Nakamura et al. [32] recently developed MERLIN method based on DRAG that can scan all discords within a specified length range. The MERLIN method finds discords of all lengths by calling the fixed-length DRAG method with all lengths within a specified length range. Although $m^{th}$ discord definition addresses the limitation of simple discord, $m$ is a user-define parameter and is challenging to set. The performance of the method is very sensitive to the variation of the length, and setting a value larger or smaller than the correct one can lead to false positive results. In contrast to these methods, our proposed method identifies abnormal subsequences based on a new definition. We define abnormal sequences as those with the lowest path weight in the graph representing the time series. Our graph model can capture both single and recurrent anomalies and normal patterns in time series.

**B. Time series graph representation for anomaly detection.** Phase space reconstruction is an effective method used to analyze non-linear time series. It involves transforming a time series [33, 34] into a set of vectors, which are then used for constructing complex networks [35]. The complex network contains an underlying complex and irregular structure of time series and can be analyzed by quantifying the graph features, such as node degree distributions and path lengths. Time series analysis using complex network has been applied to various domains [36, 37]. In [37], a new method was developed to map EEG time series to a complex network. Then, the network is used to extract sudden fluctuations (anomalies) in EEG time series. There are some graph-based outlier detection methods that map a time series to a graph by discovering relationships [38]. A method based on a time series graph representation is developed in [39] to detect outliers. They applied a sliding window and calculated the distance between subsequences. Then, they use each subsequence as a node, and the weights of the edges are distances. They developed a node clustering model based on graph to detect outliers. Although these techniques are similar to our proposed method as they apply a graph representation to approach the problem, our graph representation of time series is more compact and organized better. Our method considers a group of subsequences as a node, while the above methods consider each subsequence as a node. A graph-based method, Series2Graph, is developed in [22] also to discover both single and recurrent anomalies; this is most related to our method.

Our GraphTS method comprises two innovative techniques when compared to Series2-Graph. First, in our GraphTS, a 2Dviz method is developed to map normalized subsequences into a 2D space by keeping the structural similarities between subsequences; that is, subsequences with similar shapes are projected into the same area in the 2D space. In contrast, in Series2Graph, the local convolution of subsequences is mapped into 2D space to enhance the wave shapes by reducing noise. Second, in our GraphTS, the 2Dviz plot space is divided into grid cells, and each cell is considered as a node to represent a group of sequences with the same shapes (i.e., structure/wave patterns). As a result, the number of nodes in the represented graph is kept below a certain number, enabling our method to be scalable and efficient when applied to large datasets. In contrast, Series2Graph considers the most crossed areas in the 2D space as a node, where repeat subsequences most likely pass through. The number of nodes in represented graph built with Series2Graph increase with increasing the size of time series data which makes more complex graph for long time series.

## 3 The problem modeling

In this paper, we introduce a novel approach to transform a time series into a graph representation. Here, each subsequence is assigned a score, transforming the problem of abnormal subsequence detection into the task of identifying subsequences with lower scores. We formally model the problem as follows.

First, we aim to detect abnormal subsequences (i.e., anomalous patterns in local regions of a time series).

*Definition 1* (Time series) A time series $T \in \mathbb{R}^n$ is a sequence of real-valued numbers $t_i \in \mathbb{R}[t_1, t_2, ..., t_n], n = | T |$ is the length of $T$ [22].

*Definition 2* (Subsequence) subsequence $T_{i,\ell} \in \mathbb{R}^n$ of a time series $T$ is a subset of continuous values on $T$ of length $\ell$ starting at position $i$; formally, $T_{i,\ell} = [t_i, t_{i+1}, ..., t_{i+\ell-1}]$ [22].

Then, we borrow the idea of a directed graph (*Definition 3*) to define a new concept, time series graph representation, in *Definition 4*.

*Definition 3* (A directed graph) A directed graph $G$ is a pair *(V,E)*, where $V$ is a finite set of nodes, and $E$ is a finite set of edges [22]. The elements of $E$ are ordered pairs of node with edge weight set $W$.

*Definition 4* (Time series graph representation, *G*) Given a time series $T$, *G(V, E)* is a directed graph that represents both recurrent and rare patterns in $T$. The $G$ is a directed graph consists of a node set $V = [v_1, v_2, ..., v_m]$, a edge set $E \subseteq \{(v_i, v_j) \mid v_i, v_j \in V\}$ and a edge weight set $W \subseteq \{w_{ij}(v_i, v_j) \mid v_i, v_j \in V\}$ [22].

Our vision is to represent a time series, $T$, as a directed graph, *G(V, E)*, which characterizes both normal and abnormal patterns in time series. In $G$, node set $V$ represents various subsequence patterns in time series, and edge set E represents the number of successive occurrences of these patterns. Edge weight $w_{ij}(v_i, v_j)$ presents the sum of successive occurrences between $v_i$ and $v_j$ patterns. Therefore, recurrent consecutive normal patterns and rare abnormal patterns in time series can be represented by paths in $G$ that are composed of high weighted edges and low weighted edges, respectively. This is based on the fact that we assume the number of abnormal patterns are less than the number of normal patterns in time series. Thus, we develop new concept of the subsequence score, which can be calculated by Eq 1 in *Definition 5* as a function of its representation path in the $G$ and can be used to rank the subsequences.

*Definition 5* (Subsequence score) We assume that *G(V, E)* is the graph representation of a time series $T$, all subsequences of length $\ell$ in $T$ and their representation paths $P_\ell = \{P_\ell(i) = < v^{i+1}, v^{i+2}, ..., v^{i+\ell} >, v \in V \text{ and } i \in [0, n - \ell + 1]\}$ in $G$. $P_\ell(i)$ is the path between the nodes $v^i$ to $v^{i+\ell}$. Then, we develop Eq 1 to calculate the subsequence *score* as follows:

$$score(P_\ell(i)) = \frac{\sum_{k=i}^{i+\ell-1} w(v^k, v^{k+1})}{\ell} . \tag{1}$$

where $w(v^k, v^{k+1})$ is the edge weight between nodes $v^k$ and $v^{k+1}$, and $\ell$ is the subsequence length.

Based on the above definitions, the problem of this paper is modeled as follows.

*The Problem Modeling.* Given a time series $T$ and subsequence length of $\ell$, we construct graph *G(V, E)* from $T$ in an unsupervised way (without knowing the labels of the subsequences in $T$). Based on the graph $G$, we calculate subsequences scores (*Definition 5*) and change the problem of detecting anomalous subsequences from time series into the problem of finding those subsequences paths in Garph $G$ that have a much lower score compared to recurrent normal subsequences.

Note that by the (*Definition 5*), the *Score* of trivial matches [3] where subsequences largely overlaps with themselves are very close to each other (e.g., the $score(P_\ell(i))$ and $score(P_\ell(i + 1))$

**Table 1. Table of symbols.**

| Symbol | Description |
|---|---|
| $T$ | a time series |
| $|T|$ | cardinality of $T$ |
| $T_{i,\ell}$ | subsequence of length $\ell$ starting at position $i$ |
| $w_g$ | input window length |
| $\ell$ | anomaly length |
| $l_{np}$ | normal pattern length |
| $Z$ | matrix of all extracted subsequences |
| $Z_n$ | normalized subsequences matrix |
| 2DSTS | reduced 2D matrix of $Z_n$ |
| $n_{cell}$ | number of grid size |
| $V$ | node set |
| $E$ | edge set |
| $W$ | edge weight set |
| $G(V, E)$ | directed graph representing $T$ |
| $P_T$ | path of time series $T$ in graph $G$ |
| $P_\ell(i)$ | path of subsequence $T_{i,\ell}$ into $P_T$ |

for subsequences $T_{i,\ell}$ and $T_{i+1,\ell}$ are almost the same as they are overlapped and only have one point difference). To avoid these trivial matches, we incorporate an "exclusion-zone" of length $\ell$ before and after the location of the subsequence to be ignored. Therefore, we exclude the trivial matches to make sure overlapping subsequences are not reported. The symbols we use in this paper are defined Table 1.

# 4 The proposed approach

In this section, we provide an overview of the GraphTS method for subsequence anomaly detection in Table 2, which summarizes all steps in our approach to detect subsequence anomalies using a graph representation of time series.

In GraphTS as shown in Table 2, the window length, $w_g$, is a user-defined parameter and is different from the length of an interesting anomaly subsequence, $\ell$. However, we set $w_g$ based on the length of normal patterns in time series. In the experimental evaluation, we show that GraphTS is robust to different values of $w_g$ when the selected value is close to the length of

**Table 2. Overview of the proposed method.**

| The GraphTS Method. |
|---|
| **input**: time series $T$, anomaly length $\ell$, input window length $w_g$ |
| **output**: subsequence anomalies |
| **Step 1 2D visualization of time series (Algorithm 1 2Dviz)**. Transfer all subsequences of length $w_g$ in $T$ into a 2D spatial-temporal space, where subsequence with similar patterns are projected into similar spatial locations; |
| **Step 2 Construction of graph (Algorithm 2 ConGraph)**. Construct a directed graph based on the 2D spatial-temporal space where spatial information is used to create the node set and temporal information is used to extract the edge set. The nodes represent the various subsequence patterns of length $w_g$ in time series and edges represent the number of successive occurrences of these patterns; |
| **Step 3 Subsequence anomaly detection (Algorithm 3 AnomalyScore)**. Calculate the abnormality score for each subsequence of length $\ell$ based on their path in the constructed graph and return a ranked list of abnormal subsequences in $T$. |

normal patterns in the time series. Although the length of the abnormal subsequence, $\ell$, can be defined by users, the proposed method is robust to accurately detect anomalous subsequences under various values of $\ell$. The remaining parameters in GraphTS are internal and can be set to a default value. For example, the number of nodes in the graph, $n_{cell}$, is set to 100 nodes. Fig 2 illustrates the procedure of our proposed method. In the following subsection, we provide details of each step in the GraphTS method as shown in Table 2.

## 4.1 2D visualization of time series

We first develop the 2Dviz algorithm (Algorithm 1) for transferring a time series into a 2-dimensional spatial-temporal space (2DSTS), where the patterns of time series subsequences are preserved. We borrow the idea from [40] to develop the 2D Visualization method. However, our method is different from [40]; in [40], the whole time series is normalized using unity-based normalization to set its value into range [0, 1] while our method utilizes *Z-normalization* (i.e., normalizing every value in a dataset such that the mean of all of the values is 0 and the standard deviation is 1) for each subsequence of time series.

In Algorithm 1, the 2DSTS is obtained via three steps: (1) subsequence extraction, (2) subsequence normalization, and (3) dimension reduction. We first extract all subsequences of length $w_g$ from $T$ at Lines 1–2 in Algorithm 1, using a sliding window with a step of 1 point and create matrix $Z \in \mathbb{R}^{(|T|-w_g+1) \times w_g}$, containing all subsequences $\{T_{i,w_g}, i \in [0, |T| - w_g + 1]\}$. Each row of matrix $Z$ is a vector of size $w_g$ and defined as $Z[i, :]$ which correspond to extracted subsequence $T_{i,w_g}$. Then, we use *Z-normalization* to bring the mean of each subsequence to zero and its standard deviation to one to enable comparison of subsequences structural similarities at Line 3. This is done by subtracting each subsequence mean $\mu_i$ from each subsequence $Z[i, :]$ and dividing it by its standard deviation $\delta_i$. We denote the normalized subsequence matrix as $Z_n$. The matrix, $Z_n$, is in high-dimensional space; that is, each data point represents a normalized subsequence that occurs at a different time interval. To reduce the dimensionality of matrix $Z_n$ to two dimensions, we utilize a Principal Component Analysis (PCA) and only the top two components are kept in a reduced 2D matrix denoted as 2DSTS.

**Algorithm 1** 2Dviz

```
input: Time series T, input length w_g
output: 2D spatial-temporal space (2DSTS)
1. foreach i ∈ [0, |T| - w_g + 1] do
2.   Z[i,:] ← T_{i,w_g};              ▷ Subsequence extraction
3.   Z_n[i,:] ← (Z[i,:]-μ_i)/δ_i;     ▷ Subsequence normalization
4. 2DSTS ← PCA.fit_tranform(Z_n);     ▷ Dimension reduction
```

Fig 3 depicts the 2DSTS for the example time series shown in Fig 2(a) by setting $w_g = 80$. Each data point represents a normalized subsequence (spatial representation) and links between two points indicate the temporal order of subsequences (temporal representation). It can be seen from Fig 3 that subsequences with the same patterns appear close to each other in 2DSTS visualization (Abnormal subsequences: $T_1 - T_4$ and normal subsequences: $T_5 - T_8$). As the normal subsequences are more than abnormal subsequences, they create denser clusters in 2DSTS visualization, because they appear more frequently in time series. Fig 4 shows the temporal patterns (trajectories) in 2DSTS visualization for normal subsequences in Fig 4(b) and abnormal subsequences in Fig 4(c). The pattern (trajectory) difference between normal subsequences ($N_1 - N_4$) and abnormal subsequences ($A_1 - A_4$) is distinguishable in 2DSTS. We utilize these spatial and temporal characteristics of 2DSTS visualization in constructing a graph.

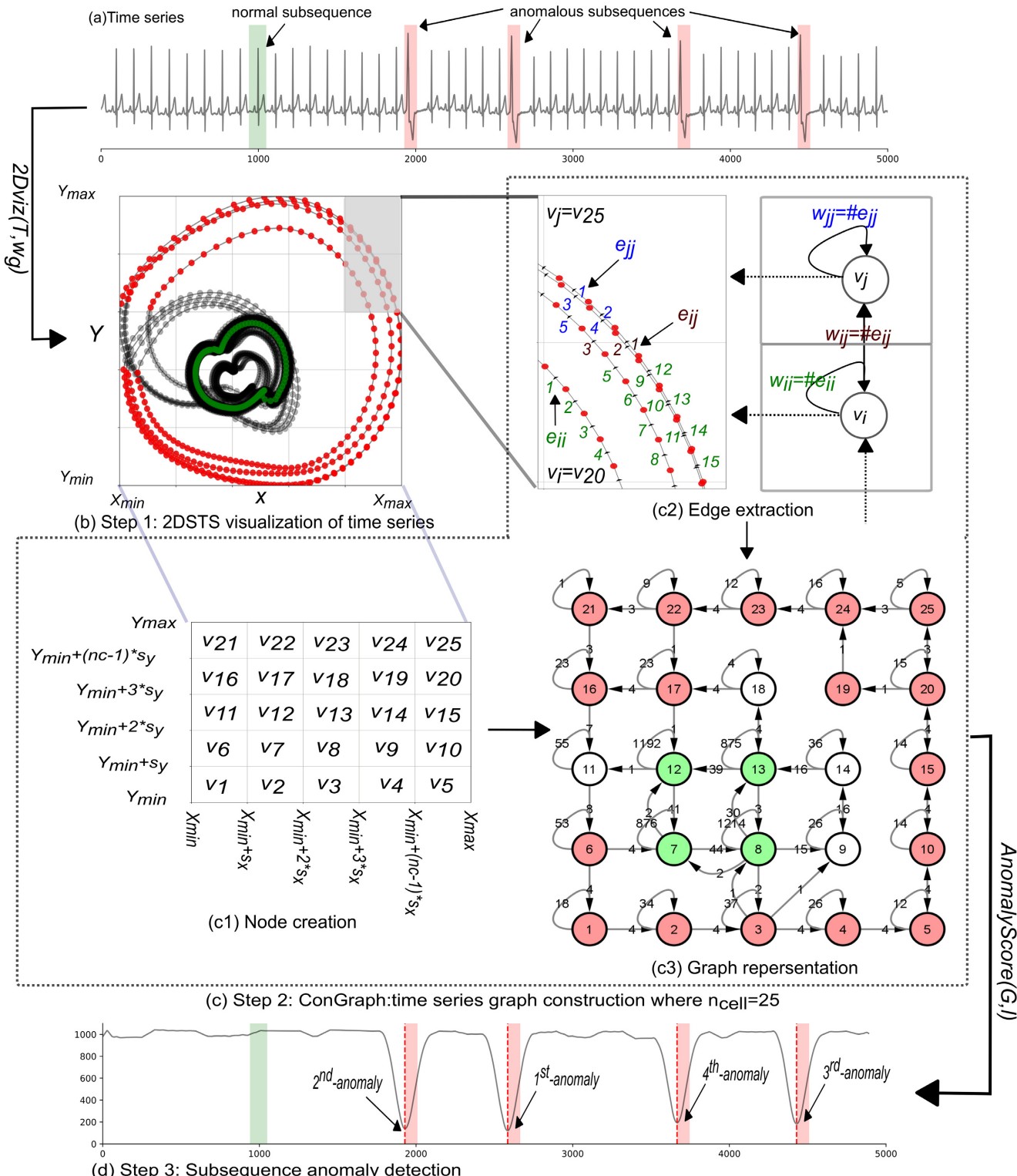

**Fig 2. Procedure of the proposed graph-based subsequence anomaly detection (GraphTS).** (a) An example time series $T$ extracted from MITBIH 1 dataset, with four anomalous subsequences (highlighted in red areas). (b) 2DSTS visualization of time series (step 1). (c) Graph construction (step 2). (d) Subsequence anomaly detection using subsequence score curves for all subsequences of $T$: low score indicates anomalous subsequences (step 3).

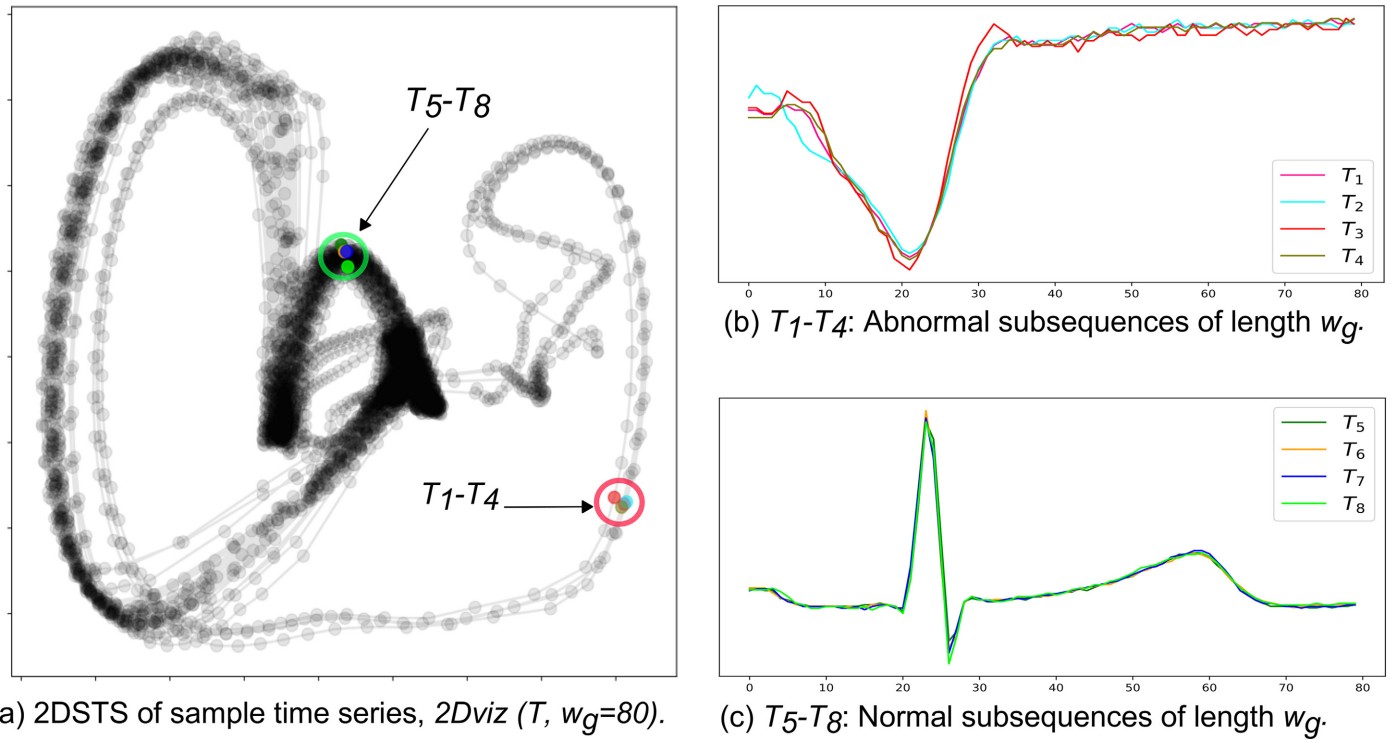

(a) 2DSTS of sample time series, *2Dviz (T, $w_g$=80)*.

(b) $T_1$-$T_4$: Abnormal subsequences of length $w_g$.

(c) $T_5$-$T_8$: Normal subsequences of length $w_g$.

**Fig 3. 2DSTS visualization.** (a) 2DSTS visualization of the sample time series shown in Fig 2(a). (b) Abnormal subsequences correspond to premature ventricular contraction (PVC) heartbeats. (c) Normal subsequences correspond to normal heartbeats.

## 4.2 Construction of graph

This step aims to create a directed graph *G(V, E)* based on the 2DSTS as shown in Algorithm 2 in order to extract abnormal and normal subsequence patterns. The main idea is to use spatial and temporal information in 2DSTS to create node set *V* and edge set *E*, respectively.

The *ConGraph* algorithm (Algorithm 2) consists of three steps: node creation (creating node set *V*), edge extraction (extracting edge set *E*), and graph construction(constructing graph representation of time series). In node creation step, the 2DSTS space is divided into $n_{cell}$ grid cells. Then, we consider each grid cell as a node $v_i \in V$. So all points in each cell (i.e., subsequences with similar patterns) are mapped to one node. As an example shown in Fig 2 (c1), we divide the 2DSTS space into $n_{cell}$=25 grid cells and create a node set, $V = [v_1, v_2, \ldots, v_{25}]$.

In edge extraction step, a directed link $(v_i, v_j)$, is established from node $v_i$ to node $v_j$ if two consecutive points (subsequences) occur between two cells in 2DSTS. This process applies to the entire 2DSTS matrix from the first point to the last point. The edge weight, $w_{ij}$, between two nodes, $v_i$ and $v_j$, is the number of times two consecutive points occur between two cells in 2DSTS. For example in Fig 2(c2), there are four times two consecutive points occur between node $v_4$ and $v_5$ (that is, four links $e_{ij}$ labeled as 1, 2, 3 and 4 in brown color), so the edge weight between these two nodes is 4 (#$e_{ij}$). We also consider self-loops where $i = j$ to map recurrent consecutive subsequences into a high weighted edge. A self-loop is established if two consecutive points appear in the same cell (node). The self-loop weight is the number of times that two consecutive points that appear in the same cell (node), e.g., defined as #$e_{ii}$ and #$e_{jj}$ in Fig 2(c2).

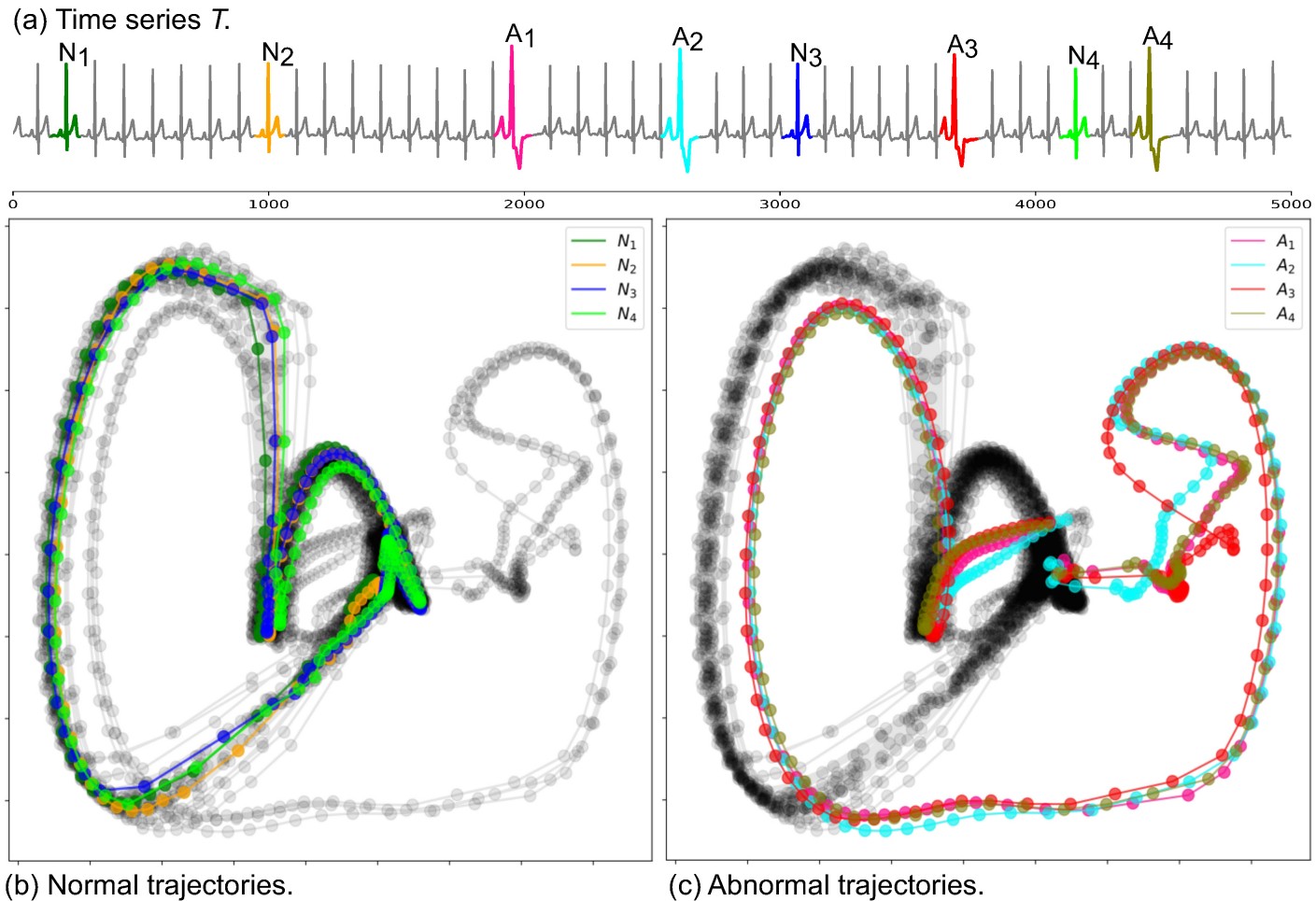

**Fig 4. Temporal patterns in 2DSTS visualization.** (a) The sample time series with four normal subsequences and four abnormal subsequences annotated as $N_1 - N_4$ and $A_1 - A_4$, respectively. (b) Normal trajectories correspond to temporal order of the normal subsequences. (c) Abnormal trajectories correspond to temporal order of the abnormal subsequences. The highlights in the time series with the corresponding highlights points in the 2DSTS space indicate noticeable differences in normal and abnormal trajectories.

For instance, the self-loops for nodes $v_4$ and $v_5$ are 6 (#$e_{ii}$ links) and 9 (#$e_{jj}$ links), respectively as shown in Fig 2(c2). In the graph construction step, the graph is constructed using node set $V$ and edge set $E$.

**Algorithm 2** ConGraph

```
input: 2-dimensional spatial-temporal space 2DSTS, n_cell=n_c × n_c
output: G(V, E)
▷ Node creation
1. Node set (V): 2DSTS is divided into grid cells n_cell by dividing
both dimension X and Y into nc boundary using s_x = (x_max−x_min)/n_c, and
s_y = (y_max−y_min)/n_c, each grid cell is represented by a node v_i ∈ V (as shown in
Fig 2(c1)).
▷ Edge extraction
2. Edge set (E): a directed link (v_i, v_j) is established from node v_i
to node v_j if two consecutive sequences occur between two nodes and its
edge weight equals the number of times that two consecutive sequences
```

```
occur between two nodes (shown as e_ij in Fig 2(c2)). a self-loop link
is also considered when two consecutive sequences appear in the same
cell (shown as e_ii and e_jj in Fig 2(c2)).
▷ Graph creation
3. Construct the graph G(V, E) using node set V and edge set E (as
shown in Fig 2(c3)).
```

The steps of the *ConGraph* algorithm are also illustrated in Fig 2(c). Note that the creation of the graph only requires one parameter, grid size $n_{cell}=n_c \times n_c$. Increasing the value of $n_{cell}$ will impact the data size of storing the graph. Decreasing the value of $n_{cell}$ will speed up the graph creation. However, it may result in loss of information regarding normal and abnormal patterns. In experimental evaluation, we demonstrate how we can find an optimal value of $n_{cell}$. Fig 2(c3) shows a *G(V, E)* graph with size of $n_{cell}=25$ (5×5). Nodes with high self-loop values correspond to normal patterns, while nodes with low self-loops values correspond to abnormal patterns.

## 4.3 Subsequence anomaly detection

In this subsection, we detail how we use the information in the generated graph, *G(V, E)*, to calculate the anomaly score for each subsequence of length $\ell$ ($T_{i,\ell}$) and detect abnormal subsequences as shown in Algorithm 3.

**Algorithm 3** AnomalyScore

```
input: G(V, E), time series T, input length ℓ
output: abnormal subsequence
▷ Transfer time series T = [t_0, t_1, ..., t_n] to a path P_T = < v^0, v^1, ...,
v^{n-w}_g+1 > in G(V, E)
1. P_T ← < >;
2. foreach i ∈ [0, n - w_g + 1] do
3.     v^i ∈ V ← T_{i,w_g};
4.     add v^i in P_T;
▷ Map each subsequence of length ℓ into path sequence P_T and calculate
its score
5. foreach i ∈ [0, n - w_g - ℓ + 1] do
6.     P_ℓ(i) = < v^i, v^{i+1}, ..., v^{i+ℓ-1} >, v ∈ V ← T_{i,ℓ};
7.     score(P_ℓ(i)) ← (∑_{k=i}^{i+ℓ-1} w(v^k, v^{k+1})) / ℓ;
8. Score(i) ← movingAve(score, w_g);
9. Anomalies ← DetectAnomaly(Score, k);
```

In Algorithm 3, we first transfer time series $T$ into a path, $P_T$, using the generated graph $G$ (lines 1–4). The path, $P_T$, is a sequence of nodes, where each node, $v^i \in V$, represents a subsequence, $T_{i,w_g}$, extracted from time series $T$. This is done by mapping all subsequences, $\{T_{i,w_g}, i \in [0, |T| - w_g + 1]\}$, to their corresponding nodes in $G$.

Fig 5(a) shows the path sequence, $P_T$, for the sample time series in Fig 2(a). The $P_T$ shows that the abnormal paths (highlighted in red) are easily distinguishable from normal path (highlighted in green). We are interested in finding abnormal sequences of length $\ell$. Therefore, each subsequence of $T$ with a length of $\ell$ ($T_{i,\ell}$) is mapped into a path, $P_\ell(i)$, using path sequence $P_T$ (lines 5 and 6). Fig 5(b) and 5(c) illustrate normal and abnormal path sequences corresponding to normal and abnormal subsequences of length $\ell = 110$, respectively. As we mentioned before, the abnormal patterns in time series are mapped into the paths in $G$ that have low weighted edges, and the normal patterns in time series are mapped into the paths in $G$ that have high weighted edges (As shown in Fig 5(d) and 5(e). For instance, the normal path sequence for the normal subsequence (starting at position $i=950$) is $P_\ell(i = 950) = < v_{14}^0, v_9^7, v_{14}^{11}, v_{13}^{18}, v_{18}^{21}, v_{13}^{29}, v_7^{36}, v_8^{43}, v_{12}^{45}, v_{13}^{48}, v_{18}^{49}, v_{23}^{50}, v_{22}^{51}, v_{21}^{56}, v_{16}^{61}, v_{11}^{64}, v_6^{67}, v_1^{70}, v_2^{77}, v_8^{80}, v_{13}^{86}, v_{14}^{107} >$ (the

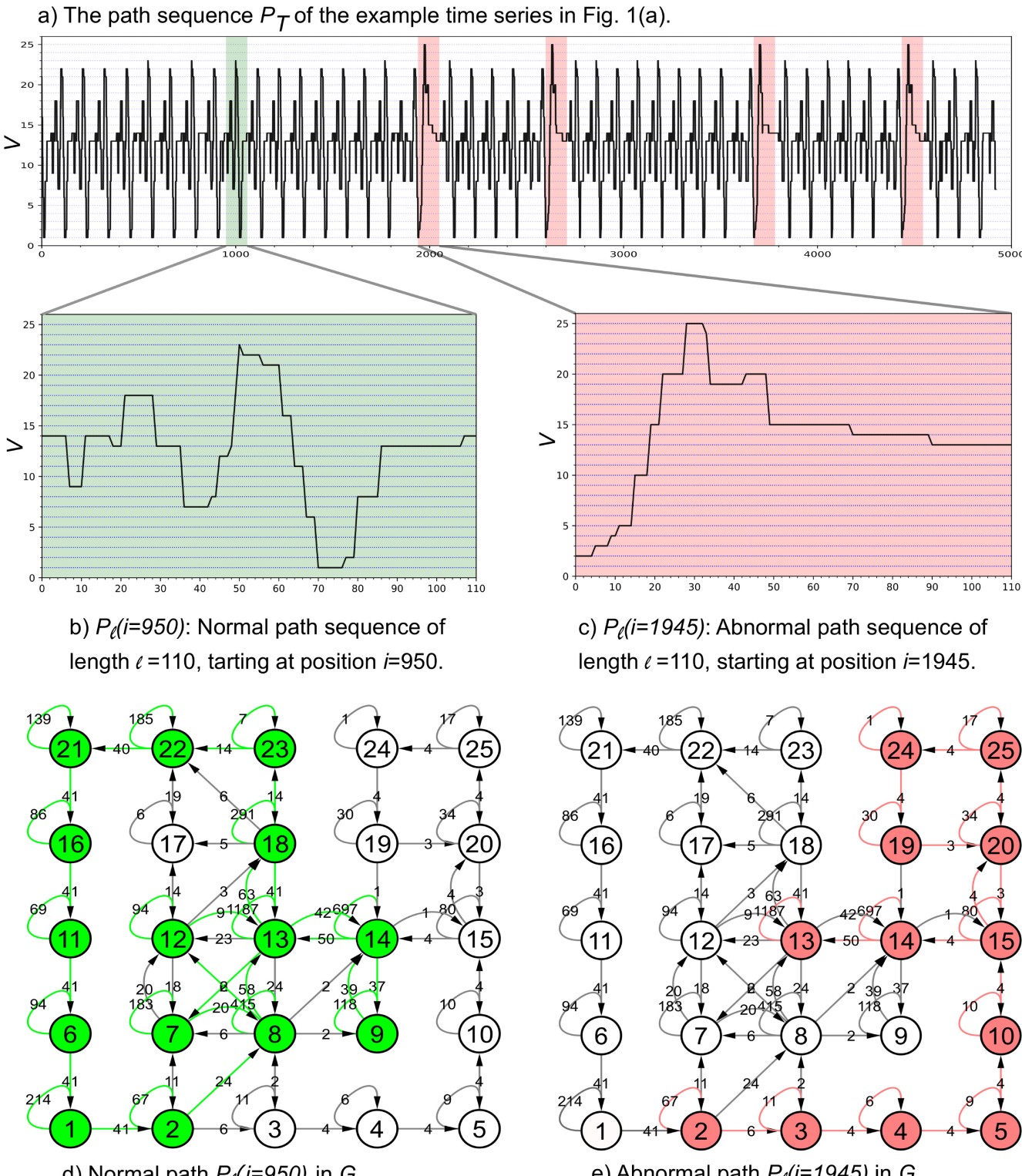

a) The path sequence $P_T$ of the example time series in Fig. 1(a).

b) $P_\ell(i=950)$: Normal path sequence of length $\ell$ =110, tarting at position $i$=950.

c) $P_\ell(i=1945)$: Abnormal path sequence of length $\ell$ =110, starting at position $i$=1945.

d) Normal path $P_\ell(i=950)$ in $G$.

e) Abnormal path $P_\ell(i=1945)$ in $G$.

**Fig 5. Normal and abnormal path sequences and their corresponding paths in graph G.** (a) Path sequence $P_T$ for the sample time series in Fig 2(a). The red and green regions are the corresponding sequence paths for abnormal and normal subsequences, respectively. (b) and (c) normal and abnormal path sequences corresponding to normal and abnormal subsequences of length $\ell$ = 110. Path in $G$:(d) normal path and (e) abnormal path.

superscript following each node denotes the first occurrence of the node in the path sequence) and its path in $G$ is shown in Fig 5(d). While the abnormal path sequence for the abnormal subsequence (starting at position $i$=1945) is $P_\ell(i = 1945) =<$ $v_2^0, v_3^5, v_4^9, v_5^{11}, v_{10}^{15}, v_{15}^{19}, v_{20}^{22}, v_{25}^{28}, v_{24}^{33}, v_{19}^{34}, v_{20}^{43}, v_{15}^{49}, v_{14}^{70}, v_{13}^{90} >$ and its path in $G$ is shown in Fig 5(e). Thus, the path weight of the abnormal path is much smaller than the path weight of the normal path. The path weight is the sum of the weights of the edges on that path.

We use this information to calculate the anomaly score for each subsequence based on its path in $G$. The score for each subsequence is defined as the average path weight for each subsequence path of length $\ell$ in $G$ and is calculated by dividing the path weight by path length $\ell$ (line 7). Then, a moving average filter is applied to the *score* vector to make sure that the score for highly overlapping subsequences has a relatively similar score (line 8). In the final step, we rank subsequences based on their scores (from the lowest to highest score) and report an anomaly list (rank, subsequence), which can be used to detect Top-$K$ abnormal subsequences (line 9). The $K$ subsequences of time series $T$ with the lowest score are considered abnormal subsequences.

Fig 6(a) shows the subsequence's *Score (i)* for sample time series in Fig 2(a). As we expect, the score for abnormal paths is much lower than the score for normal path. For instance, the scores for the normal path $P_\ell(i = 950)$ and the abnormal path $P_\ell(i = 1945)$ (Shown in Fig 5(d) and 5(e)) are 465.29 and 227.77, respectively. We exclude the trivial matches by incorporating "exclusion-zone" (shown as gray areas in Fig 6(a)) of length $\ell$ before and after the location of each lowest score to avoid reporting overlapping subsequences. Considering $K$=4 for detecting four abnormal subsequences in the sample time series, the GraphTS reports four abnormal subsequences with the lowest scores, starting at different positions at $i$=3685, 4435, 1945, and 2594 and ranks them based on their scores as $1^{st}$, $2^{nd}$, $3^{rd}$ and $4^{th}$, respectively. Fig 6(b) illustrates the four abnormal subsequences that were correctly detected by our GraphTS method.

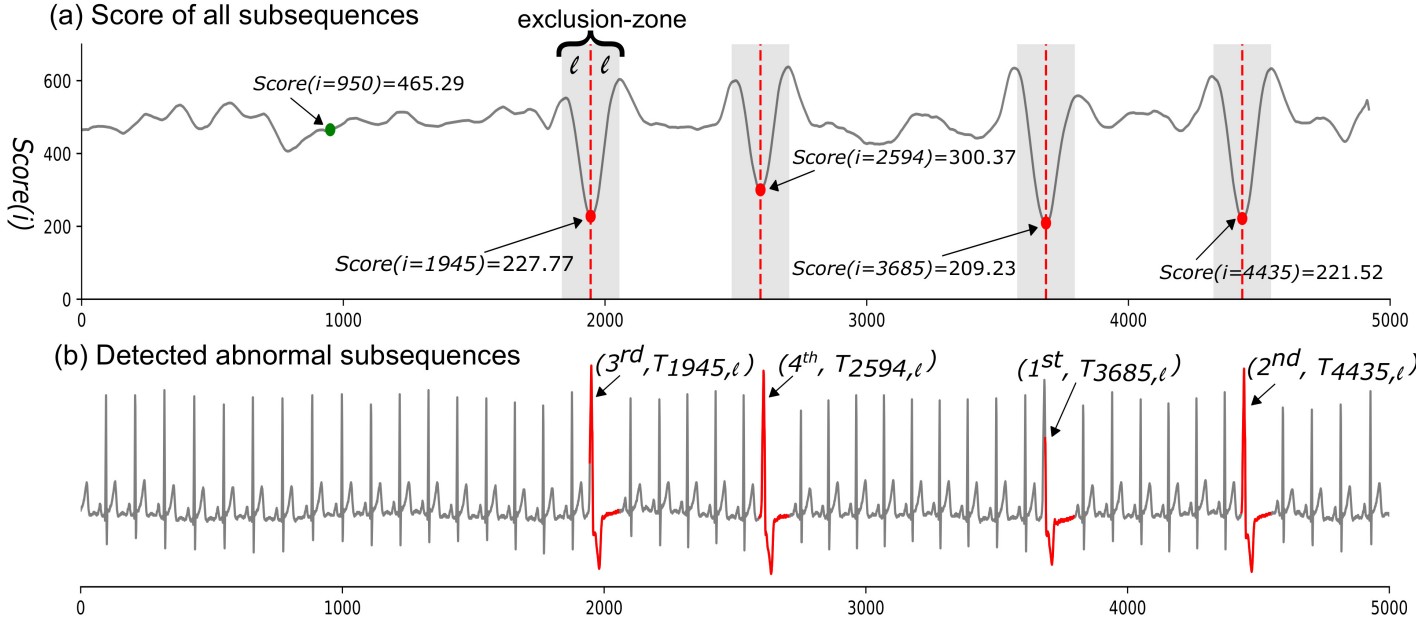

**Fig 6. Subsequence anomaly detection.** (a) Score of each subsequence in the sample time series in Fig 2(a). The red points indicate the four lowest scores corresponding to four subsequences with the length of $\ell$ = 110 starting at different positions $i$ in time series. The exclusion-zones are highlighted in gray. (b) Top-4 abnormal subsequences reported and ranked ($1^{st} - 4^{th}$) by the GraphTS.

We note that the selection of value for parameter $K$ is not necessary as our method can provide a ranked list for all subsequences. Moreover, the generated Graph model can be used to discover anomalies with various lengths. From Algorithm 3 (line 5–9), we only need to map subsequences with various lengths $\ell \in [minL, maxL]$ into path sequence $P_T$ and calculate their score to find anomalies with different lengths. Therefore, our proposed method can identify anomalies of different lengths much faster than methods that need to run for different lengths to detect variable length anomalies.

## 5 Experimental study

In this section, we have conducted extensive experiments to evaluate the accuracy and efficiency of our GraphTS method on various real-world datasets. We present the experimental setup in Section 5.1, discuss optimal parameters of GraphTS in Section 5.2, and evaluate its Top-$K$ accuracy in Section 5.3 and efficiency (execution time) in Section 5.4.

### 5.1 Experimental setup

We have implemented our GraphTS in Python. The experiments were carried out on a computer with an Intel CORE i7–8650U CPU @ 1.90GHz and 16GB memory, running a 64-bit Windows 10 operating system. To ensure the reproducibility of our experiments, we have built a webpage (https://sites.google.com/view/graphts) with the source code and datasets.

We evaluated our proposed method using real datasets from various domains and UCR benchmark as shown in Tables 3 and 4. The datasets are listed as follows.

1. Simulated engine disks data set (SED) [41, 42], which contains disk revolutions time series collected at NASA Glenn Research Center's Rotordynamics Laboratory.

2. MIT-BIH Supraventricular Arrhythmia Database (svdb) and Arrhythmia Database (mitdb) [43, 44], which consist of four electrocardiogram recordings with different arrhythmia (heart anomalies).

3. UCR Time Series Anomaly Datasets: we evaluate our proposed method on recently published benchmark dataset, the UCR Time Series Anomaly Datasets [45]. We use 100 time series (file numbers from # 109 to #208) in various domains from the UCR Time Series Anomaly Datasets [45]. We group time series in UCR benchmark datasets based on their application domains as shown in Table 4. Each dataset in UCR benchmark datasets has training and testing parts. The training part is free of anomalies while the testing part has only one anomaly or one significant anomaly if it has more than one anomaly.

Our evaluation strategy is in three steps: (1) in Section 5.2, we study the sensitivity of the proposed method on its parameters and provide the optimal values for them, and (2) in

**Table 3. Datasets used to evaluate the proposed method, with length of time series ($n$), length of anomaly ($\ell$), number of anomalies ($A$) and domain.**

| Datasets | | $n$ | $\ell$ | $A$ | Domain |
|---|---|---|---|---|---|
| 1. SED | | 100 K | 100 | 50 | Electronic |
| 2. MIT-BIH | MITBIH 1 (SAD803) | 200 K | 80 | 130 | Cardiology |
| | MITBIH 2 (SAD820) | 200 K | 100 | 159 | Cardiology |
| | MITBIH 3 (AD116) | 200 K | 200 | 32 | Cardiology |
| | MITBIH 4 (AD119) | 200 K | 250 | 125 | Cardiology |

**Table 4. List of the UCR benchmark datasets used to compare the performance of the proposed method with Series2Graph method.** Group name (*GN*), number of datasets in each group (#D), and File number considered in each group (#F).

| | GN | #D | #F |
|---|---|---|---|
| UCR datasets | ECG | 25 | 109–111,119–126,163–166,178–180,182–183,192–196 |
| | Internal bleeding (IB) | 13 | 132–144 |
| | Giat | 12 | 127–131,167–169,170–172,181 |
| | Insect | 11 | 145–150,173–177 |
| | Respiration | 8 | 184–191 |
| | CHARI | 8 | 201–208 |
| | Weather | 6 | 113–118 |
| | NASA | 5 | 156–160 |
| | Other | 12 | 112,151–155,161–162,197–200 |

Section 5.3–5.4, we evaluate the robustness of the GraphTS for anomaly detection in terms of Top-*K* accuracy and execution time using real datasets and compare it with STOMP method [21] and Series2Graph [22]. (3) in Section 5.5, we evaluate the ability of the GraphTS for detecting anomalies of various lengths using 100 time series from UCR benchmark datasets and compare it with Series2Graph [22] in terms of Top-*1* accuracy and execution time.

## 5.2 Optimal parameters of the proposed method

In this subsection, we evaluate the sensitivity of the GraphTS methods and discuss the optimal values of its three parameters: $w_g$, $n_{cell}$ and $\ell$.

**(1) Effect of $w_g$.** We first evaluate the effect of the input parameter $w_g$ of the proposed method. As this parameter is used to generate graph *G*, we ensure that the GraphTS method is robust to variation of this parameter for accurately representing the patterns (normal and abnormal) in time series; this is critical to detect anomalies accurately. Due to the fact that a time series may not contain any anomaly, we set the length of $w_g$ based on the length of normal pattern ($l_{np}$) in time series to guarantee that the generated graph can characterize the normal pattern. To evaluate the sensitivity of GraphTS to $w_g$, we measure Top-*k* accuracy, by setting *k* and $\ell$ equal to the number of anomalies, *A*, and the length of anomalies $\ell$, respectively, in each real dataset and let $n_{cell}$ =100, and then vary the length of $w_g$ based on the length of $l_{np}$ in each dataset. The $l_{np}$ length can be identified easily from each dataset using the Multi-Window-Finder method [46]. For example, the $l_{np}$ length for the ECG dataset is the heartbeat's length. Table 5 shows the $l_{np}$ values for each dataset. Fig 7 shows the stability of the proposed method by varying the length of $w_g$. The performance of the proposed method is stable when the length of $w_g$ used to create the graph are smaller than the length of $l_{np}$, it shows that selecting a value of $w_g$ smaller than the length of $l_{np}$ will lead to better accuracy. Therefore, we set $w_g = l_{np}$-20 for the rest of the experiments.

**(2) Effect of $\ell$.** We evaluate the robustness of GraphTS to the variation of subsequence length $\ell$. We measure Top-*K* accuracy, setting *k* equal to the number of anomalies (*A*) in each

**Table 5. Selected $l_{np}$ values for the SED and MITBIH datasets.**

| Dataset | SED | MITBIH 1 | MITBIH 2 | MITBIH 3 | MITBIH 4 |
|---|---|---|---|---|---|
| $l_{np}$ | 100 | 100 | 110 | 270 | 320 |

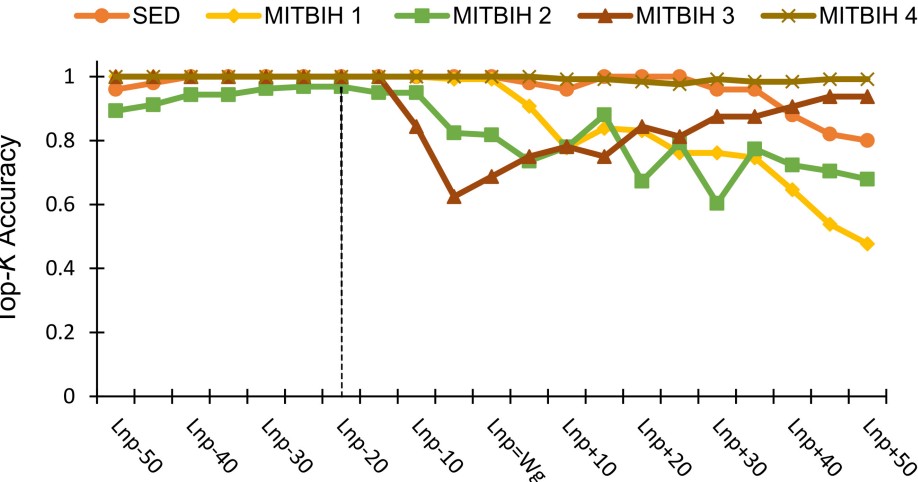

**Fig 7. GraphTS Top-$K$ accuracy vs variation on window size ($w_g$).**

real dataset and $w_g$ equal to the length of $l_{np}$-20, and then vary the length of $\ell$. We use $n_{cell}$ = 100 for this experiment. Fig 8 shows the top-$k$ accuracy of the proposed method by varying the $\ell$. These results indicated that we can identify anomalies with high accuracy by varying $\ell$ using a fixed $w_g$. Therefore, the proposed method is robust against anomaly length and does not need to know the exact length of the anomaly. Fig 8 also demonstrates that our proposed method is robust to the variable length of anomalies.

**(3) Effect of $n_{cell}$.** We evaluate the influence of node number $n_{cell}$ on the performance of GraphTS and execution time for the Graph generation. We measure Top-$k$ accuracy, setting $k$ equal to the number of anomalies ($A$) in each dataset and $w_g$ equal to the length of $l_{np}$-20 in each dataset and $\ell$ equal to the length of anomalies, and then vary the value of $n_{cell}$ from 4 to 400. Fig 9(a) illustrates the top-$k$ accuracy changing with the values of $n_{cell}$. Even though the performance of the proposed method drops for a small number of node cells ($n_{cell}$< 100), the

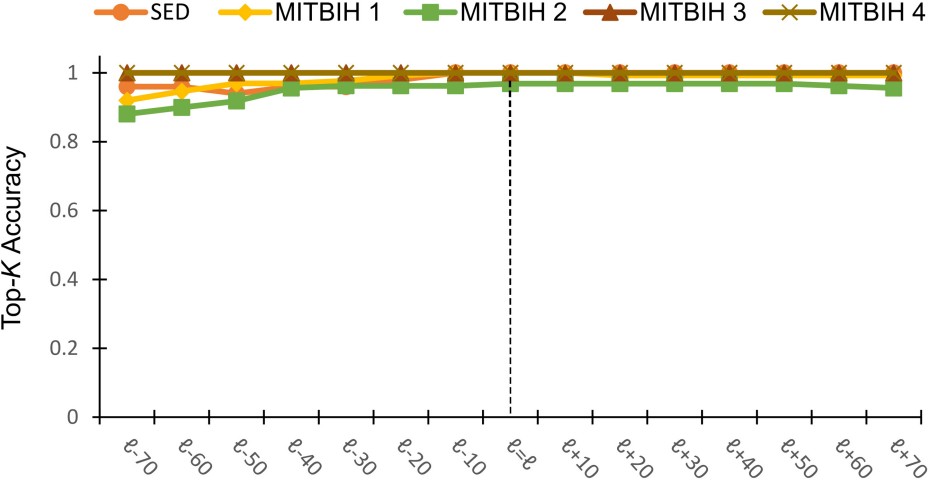

**Fig 8. Top-$K$ accuracy of GraphTS changing with $\ell$.**

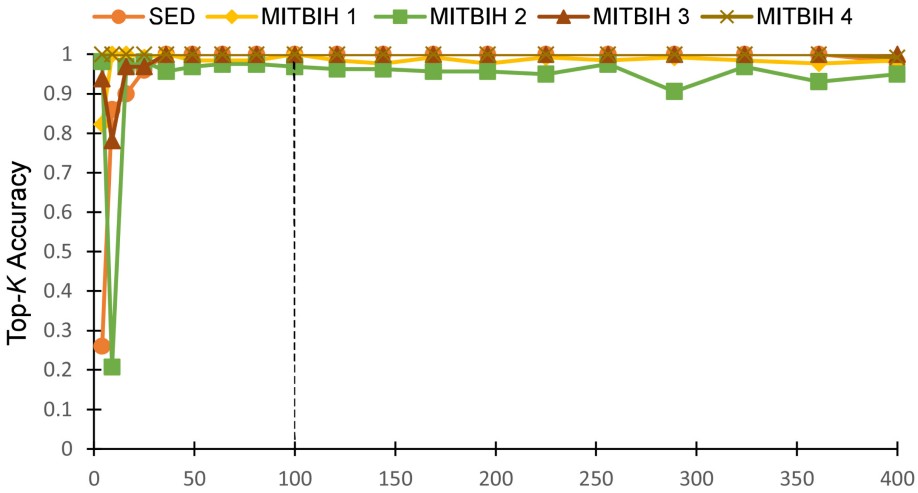

**(a)** GraphTS Top-$K$ accuracy changing with $n_{cell}$.

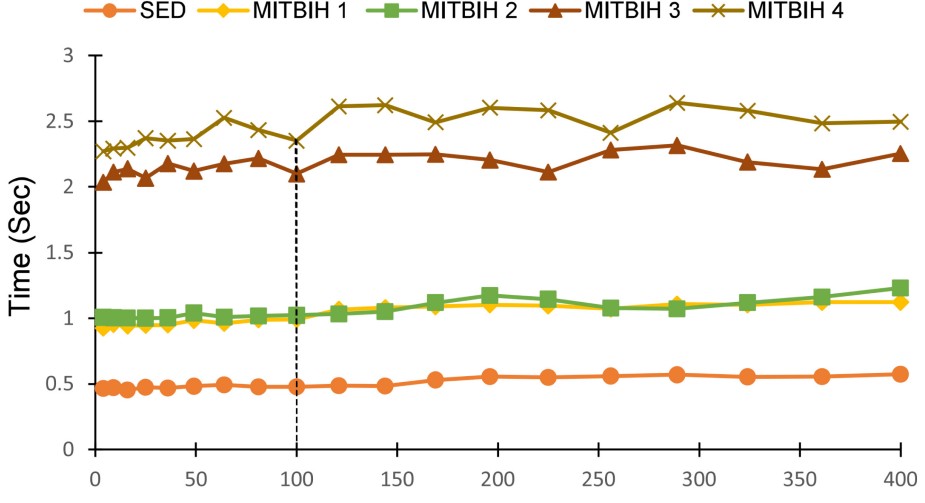

**(b)** Graph construction execution time changing with $n_{cell}$.

**Fig 9. Effect of $n_{cell}$ on GraphTS performance.**

performance remains stable when $n_{cell} \geq 100$. This indicates that just satisfying $n_{cell} \geq 100$ will yield satisfactory results. Fig 9(b) shows the execution time for graph generation versus the number of node cells ($n_{cell}$). Increasing the value of $n_{cell}$ will impact the graph size. Decreasing the $n_{cell}$ will speed up a little the graph creation. However, it may result in information loss regarding the normal and abnormal patterns. To avoid information loss and make the GraphTS scalable on large datasets, we select $n_{cell} = 100$ for the rest of the experiments.

## 5.3 Top-$k$ accuracy

In this section, we report the evaluation result based on Top-$K$ Accuracy on both SED and MIT-BIH datasets. We compare the proposed method to the counterpart STOMP and Series2-Graph methods.

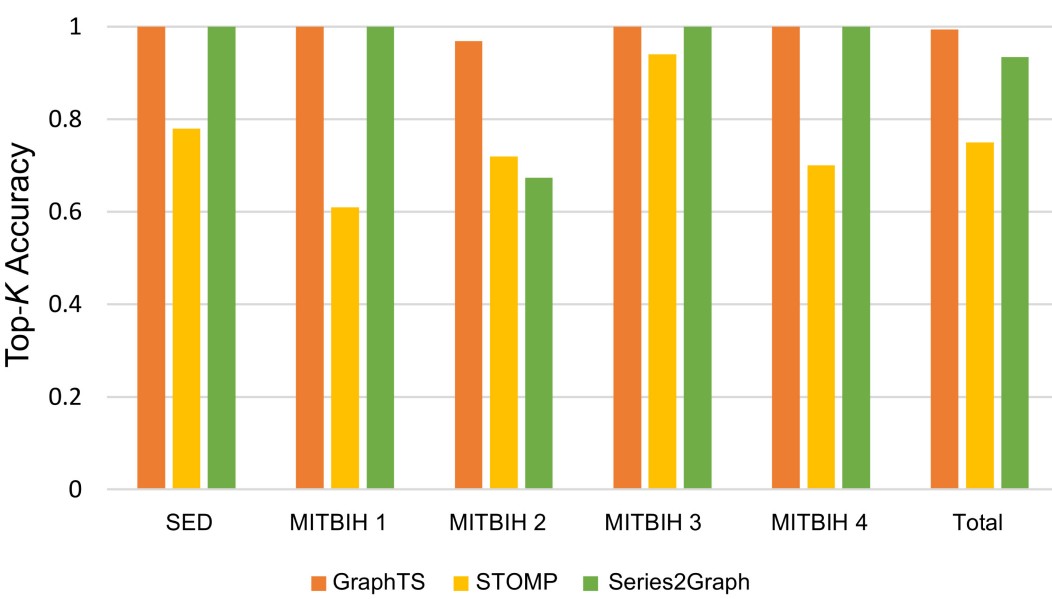

**Fig 10. Top-*K* accuracy.**

We evaluate the ability of the proposed method to correctly detect $k$ abnormal subsequences in real datasets (MIT-BIH and SED). For the GraphTS, we set $w_g = l_{np}\text{-}20$, $\ell$ equals to the length of anomaly, and $n_{cell}= 100$ and retrieve Top-$k$ anomalous subsequences. For Series2-Graph, we used the same value selected in [22] for its parameters, pattern length ($l_p$=50), and query length ($l_q$=75), for datasets SED and MITBIH1 and MITBIH2. For datasets MITBIH3 and MITBIH4, we set $l_q$= equals to the length of anomaly, and pattern length $l_p$=$2l_q$/3. Fig 10 shows the Top-$k$ accuracy for the GraphTS, STOMP and Series2Graph for each dataset. These real datasets contain multiple anomalies. The proposed method achieves perfect accuracy and outperforms both STOMP and Series2Graph methods in total accuracy. Both GraphTS and Series2Graph achieve perfect accuracy on datasets: SED, MITBIH 1, MITBIH 3 and MITBIH 4. However, the GraphTS perform much better on MITBIH 2 dataset and obtain accuracy of 96% while the Series2Graph achieve accuracy of 67%. The STOMP method achieve lower accuracy because anomalies do not relate to uncommon subsequences as abnormal sequences with a similar pattern are repeated in these datasets. These results indicate the ability of GraphTS to accurately detect recurrent anomalies.

## 5.4 Efficiency

In this subsection, we report the efficiency of the proposed method in real datasets (MIT-BIH and SED) and compare it with two counterpart methods (STOMP and Series2Graph). The results of execution time changing with dataset sizes are shown in Fig 11. We use several prefix snippets (20K, 50K, 100K, 150K, 200K points on time series) of the MITBIH datasets and use several prefix snippets (20K, 40K, 60K, 80K, 100k points) of the SED dataset. For all datasets, we set $k$ equal to the number of anoalies in each snippet. The results show that the proposed method is faster than STOMP and Series2Graph. The GraphTS is at least two orders of magnitude faster than Series2Graph and four orders of magnitude faster STOMP method. To show the scalability of GraphTS, we also report the number of nodes in each represented graph generated by GraphTs and Series2Graph. We aim to demonstrate the memory and time efficiency

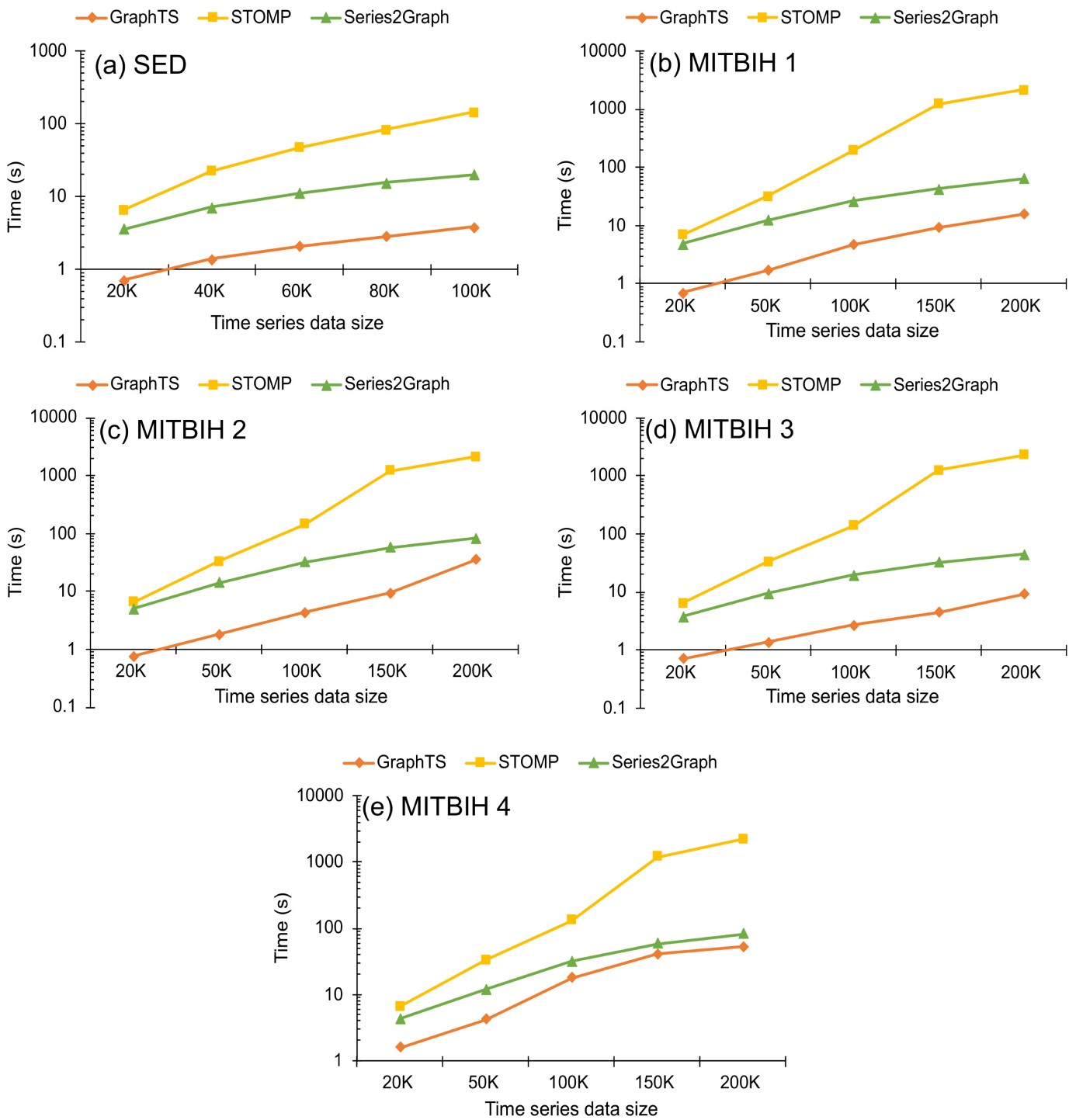

**Fig 11. Execution time changing with time series data sizes.**

of our GraphTS. It is important to note that we were unable to compare with the STOMP method as it is not a graph-based method. Fig 12 shows the number of nodes in each represented graph changing with dataset sizes. As we set $n_{cell}$ equal to 100, the number of nodes in all represented graphs are kept equal or below 100 and the change in time series data size does

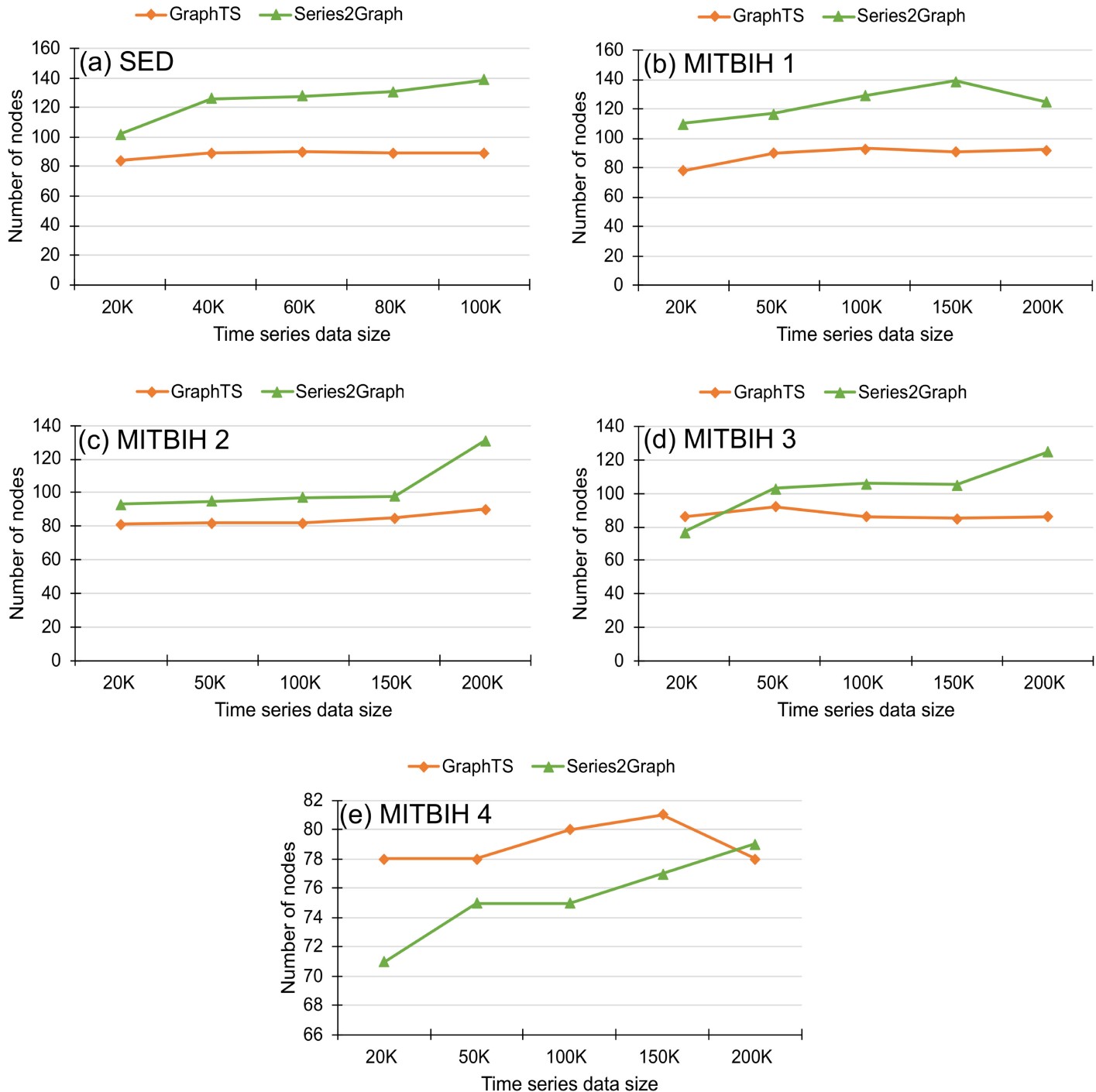

**Fig 12. Number of nodes in represented graph changing with time series data sizes.**

not affect the size of represented graph built by GraphTS. However, as the most crossed areas in the 2D space is considered as a node in Series2Graph, the number of nodes in represented graphs built by Series2Graph method increase by increasing the time series data size as shown in Fig 12. This result indicates the capability of GraphTS on representing long time series

without increasing the number of nodes in the represented graph. Therefore, it is more time and memory efficient than Series2Graph method.

## 5.5 Variable-length anomaly detection

In this section, we evaluate the ability of the proposed method to detect anomalies of various lengths. We report the effectiveness of proposed method on detecting abnormal subsequences in different domain datasets using the UCR benchmark datasets. We also compare our methods with Series2Graph in terms of accuracy and execution time.

For each time series in the UCR benchmark datasets, we only focus on the testing part of each signal and aim to identify the Top-*1* anomaly. This is because there is only one anomaly present in each testing time series. For the GraphTS, we set $n_{cell}$= 100 for all datasets, and $w_g$ is set to be smaller than the length of $l_{np}$ in each dataset to build GraphTS model (see S1 Table for selected values of $w_g$ and $\ell$ for each dataset). For the Series2Graph, we also consider the same value of $w_g$ that we use for our GraphTS method as pattern length to build Series2Graph model. Both generated GraphTS and Series2Graph models can be used to discover anomalies with various lengths. Therefore, we report top-1 anomaly result for different lengths in a given range from minimum to maximum length ($\ell \in$ [minL, maxL]) in each dataset. The minimum length (minL) is set to 10 and the maximum length (maxL) is set to max(100, $l_{np}$) for each dataset. As each dataset contains only one anomaly we consider the Top-*1* discord for all methods and report the results as binary (detected | not-detected) if the method can locate anomaly correctly for any $\ell \in$ [minL, maxL]. Then, we calculate the accuracy by dividing the number of corrected detection of each method by the total number of datasets (#D) in each group as shown in Table 4. The anomaly detection results of each methods and its execution time for each dataset are provided in S1 Table.

Fig 13 shows a summary of performance comparisons of the GrapTS with the Series2Graph method in terms of accuracy and execution time. From Fig 13(a), we can confirm that the GraphTS outperforms the Series2Graph method in terms of anomaly detection in seven groups for UCR datasets and achieves the total accuracy of 84% while Series2Graph obtains the total accuracy of 61%. Our GraphTS method outperforms the Series2Graph method in terms of execution time in all groups as shown in Fig 13(b). The total execution time results indicate that the proposed method is much faster than the counterpart Series2Graph method. The proposed GrapTS method needs only 2319s to process all datasets which is about 30 times faster than the Series2Graph. In all datasets (see S1 Table), the execution times of the proposed GrapTS method are much lower than the counterpart method. These results show the ability of the proposed GraphTS to identify anomalies with variable lengths using the same generated Graph model.

In summary, the experimental results show that the proposed GraphTS outperforms the counterparts Series2Graph and STOMP methods in terms of accuracy and execution time. The GraphTS has several merits as follow:

1) The GraphTS does not need labels for subsequences to generate graph model and can be applied in different domain.

2) The GraphTS is robust to the variation of anomaly length $\ell$ and does not need any information about the anomalous subsequence.

3) The GraphTS is able to correctly detect recurrent anomalies.

4) The GraphTS is much faster than the counterpart methods. The generated graph can be used to detect anomalies with different lengths.

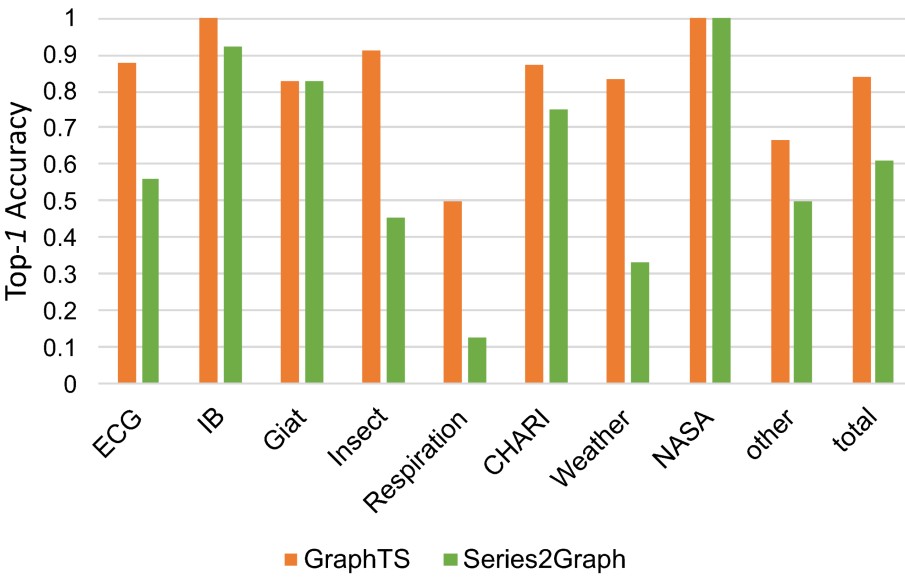

(a) Top-1 accuracy

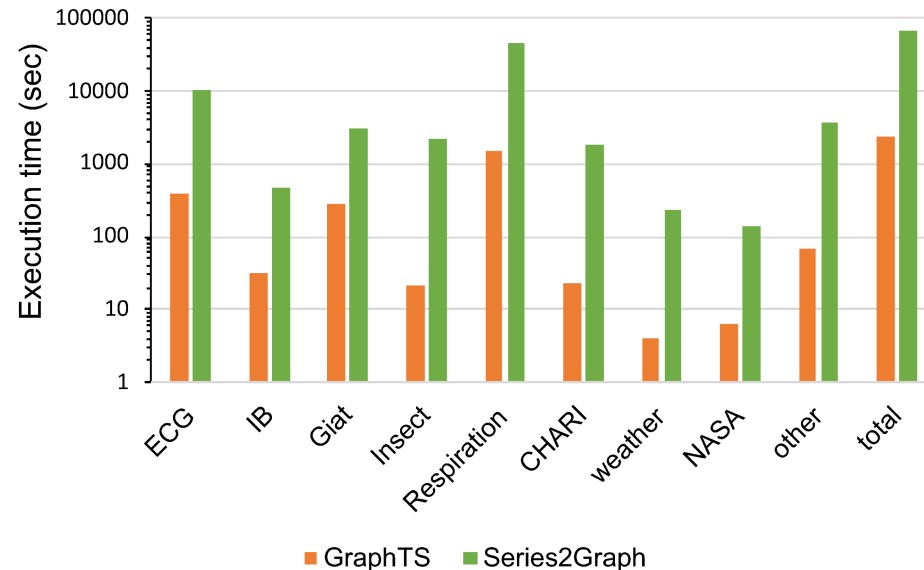

(b) Execution time

**Fig 13. Summary of performance comparisons of the proposed method with the Series2Graph method on UCR benchmark datasets.**

A limitation of the current study is that the proposed method still requires the length of normal patterns in time series to build the represented graph. However, the length can be identified from time series using methods such as Multi-Window-Finder in [46]. In future work, we aim to enhance the proposed GraphTS by incorporating additional features. Firstly, we plan to explore different time series segmentation methods such as Multi-Window-Finder in [46] to automate the process of identifying window sizes based on various behaviours in the time series and setting the input parameter $w_g$ in GraphTS. By leveraging these methods, we

can improve the efficiency and adaptability of the anomaly detection process. Furthermore, we intend to leverage the represented graph in GraphTS for time series motif discovery. Building upon the generated path sequence of the time series, we will employ techniques to identify repeated node sequences that represent normal recurring patterns (motifs) in the time series data. We can gain valuable insights into the underlying patterns and structures within the time series by capturing these recurring motifs. By incorporating these advancements, we aim further enhance the effectiveness and versatility of GraphTS, making it a more robust and comprehensive tool for anomaly detection and pattern discovery in time series data.

## 6 Conclusion

In this paper, we have presented GraphTS, a novel subsequence anomaly detection method designed to address the limitations of existing models. Our approach overcomes the challenges of needing prior knowledge of anomaly length and quantity, as well as the inability to detect recurrent anomalies. By leveraging a graph representation of time series data, GraphTS enables the efficient detection of both rare and frequent subsequence anomalies across diverse domains. The method involves embedding time series subsequences into a 2D space using our developed 2D visualization technique, followed by constructing a graph based on this representation. Notably, GraphTS does not rely on labeled data for training and is capable of detecting anomalies of varying lengths using a single represented graph. Experimental results demonstrate that GraphTS outperforms comparable methods such as STOMP and Series2-Graph in terms of accuracy and execution time.

## Supporting information

**S1 Table. Performance comparisons with the Series2Graph methods on UCR datasets.** We show the selected parameters, the anomaly detection results of each method as well as its execution time for each dataset in UCR archive. Green cells/red cells designate dataset where methods detect/not detect the anomaly in that dataset.
(PDF)

## Author Contributions

**Conceptualization:** Roozbeh Zarei, Guangyan Huang.

**Data curation:** Roozbeh Zarei, Junfeng Wu.

**Funding acquisition:** Guangyan Huang.

**Investigation:** Roozbeh Zarei.

**Methodology:** Roozbeh Zarei, Guangyan Huang.

**Software:** Roozbeh Zarei.

**Supervision:** Guangyan Huang.

**Validation:** Roozbeh Zarei, Guangyan Huang.

**Visualization:** Roozbeh Zarei, Junfeng Wu.

**Writing – review & editing:** Roozbeh Zarei, Guangyan Huang, Junfeng Wu.

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
