## [Decision Letter · Decision Letter 0]

2 Aug 2022

PONE-D-22-15666GraphTS: Graph-Represented Time Series for Subsequence Anomaly DetectionPLOS ONE

Dear Dr. Huang,

Thank you for submitting your manuscript to PLOS ONE. After careful consideration, we feel that it has merit but does not fully meet PLOS ONE’s publication criteria as it currently stands. Therefore, we invite you to submit a revised version of the manuscript that addresses the points raised during the review process.

We look forward to receiving your revised manuscript.

Kind regards,

Vijayalakshmi Kakulapati, Ph.D

Academic Editor

PLOS ONE

Journal Requirements:

Reviewers' comments:

Reviewer's Responses to Questions

**Comments to the Author**

1. Is the manuscript technically sound, and do the data support the conclusions?

Reviewer #1: Yes

Reviewer #2: Yes

2. Has the statistical analysis been performed appropriately and rigorously? 

Reviewer #1: No

Reviewer #2: N/A

3. Have the authors made all data underlying the findings in their manuscript fully available?

Reviewer #1: Yes

Reviewer #2: Yes

4. Is the manuscript presented in an intelligible fashion and written in standard English?

Reviewer #1: Yes

Reviewer #2: Yes

5. Review Comments to the Author

Reviewer #1: The above article currently has several weaknesses, which are described below.

1: the Related Work section is petite. Please add more work to this section and discuss it briefly.

2: Please provide a detailed description of your proposed model.

3: Specify the limitations and drawbacks of the proposed method.

4: It is recommended to re-examine and design the Figures.

5: A deep and detailed comparison with other methods is mandatory.

6: The authors should also clarify the motivation and main contribution of applied approach more clearly in the introduction and conclusion sections.

7: The results and discussion section has to be improved, where more details of the achieved results should be stated clearly in this section. In addition, authors also have to provide some insight discussion of the results.

Reviewer #2: This article proposes a graph-based anomaly detection method for time series data. The proposed method, GraphTS,

aims to do efficient detection of both recurrent and rare anomalies.

This article is well-organized, well-explained, and easy to read. However, I have some concerns.

Concerns:

1. Image quality is very poor. For example, it was hard to determine what's going on in (c1) and (c2) in Figure 2. The edge

labels in (C3) of the same figure is almost illegible. Please replace the current images with high quality ones. Figure 3 is so blurry that the points and links between points are hard to distinguish. Better quality images would be help to understand the method well.

2. It's confusing that Z has been represented as a vector in Algorithm 1 (but it's actually a 2D matrix)? The

symbols and notations used in the algorithm should be defined clearly.

3. The authors claim, in section 5.3, that the proposed method outperforms both STOMP and Series2Graph methods. This

doesn't really give an accurate picture of the comparison. Out of 5 datasets, the proposed method outperforms Series2Graph on only one and performs the same on the remaining datasets. The authors should revise their claim and make it more specific. The one dataset on which the proposed method performs better, it does 20% better than Series2Graph which is satisfactory. However, if we consider the overall performance comparison shown in Figure 10, the proposed method does not do significantly better than Series2Graph. It would be more appealing if the authors used more datasets and could show that their proposed method performs better on at least about 50% of the datasets used in the experiment.

However, considering the scalability factor, the proposed method does have some useful contributions.

Minor issues:

Some grammar and spelling errors exist. For example, it should be "rest", not "reset", in the paragraph under Table 1.

6. PLOS authors have the option to publish the peer review history of their article (what does this mean?). If published, this will include your full peer review and any attached files.

Reviewer #1: No

Reviewer #2: No

---

## [Author Response · Author response to Decision Letter 0]

2 Feb 2023

Response to reviewers’ comments for: GraphTS: Graph-Represented Time Series for Subsequence Anomaly Detection

We thank the editor and all the reviewers for their constructive feedback to help improve this paper. Below please find the point-to-point responses to the comments raised by the reviewers. Accordingly, we highlighted our modifications in the new version.

Response to Reviewer 1

Comment 1: the Related Work section is petite. Please add more work to this section and discuss it briefly. 

Response: 

As suggested, we have added more related works (i.e., [39] [42] [43] [44]) and discussions to the Related Work section (see page 4, lines 121-157).

Comment 2: Please provide a detailed description of your proposed model. 

Response: 

We have added more overall description of the GraphTS method in the Introduction section of the revised version (see page 2, lines 46-60). Table 2 also summarizes the three steps of the GraphTS method, and we provide an example to explain the three steps in Figure 2. Accordingly, Sections 4.1 (Algorithm 1), 4.2 (Algorithm 2) and 4.3 (Algorithm 3) have been developed in details to explain the mechanism of each step.

Comment 3: Specify the limitations and drawbacks of the proposed method. 

Response: 

We have provided the limitation and future work for the proposed method (see page 17, lines 565-571)

Comment 4: It is recommended to re-examine and design the Figures. 

Response: 

We have re-examed all figures and replaced them with high-quality ones. Particularly, we have redesigned Figure 2 to ensure all subfigures are clear and readable. 

Comment 5: A deep and detailed comparison with other methods is mandatory. 

Response: 

In the revised version, we have conducted a deep and detailed comparison with other methods (STOMP and Series2Graph) by adding more experimental results (see Figure 13 and S1 Table) on more datasets (see page 12, lines 412-419 and Table 4) and added the whole Section 5.5 for detailed explanation.

Comment 6: The authors should also clarify the motivation and main contribution of applied approach more clearly in the introduction and conclusion sections. 

Response: 

As suggested, we have revised the introduction and conclusion sections to clarify our motivation and contribution (see introduction section lines 86-103 on page 3 and conclusion section on page 17).

Comment 7: The results and discussion section has to be improved, where more details of the achieved results should be stated clearly in this section. In addition, authors also have to provide some insight discussion of the results.

Response: 

We have improved the result and discussion section as suggested. We have added more detailed results and insight in the revised paper (see Section 5.5, Figure 13 and S1 Table)

Response to Reviewer 2

Comment 1. Image quality is very poor. For example, it was hard to determine what's going on in (c1) and (c2) in Figure 2. The edge labels in (C3) of the same figure is almost illegible. Please replace the current images with high quality ones. Figure 3 is so blurry that the points and links between points are hard to distinguish. Better quality images would be help to understand the method well.

Response: 

We have re-examed Figure 3 and all of the other figures and replaced them with high-quality ones. Particularly, we have redesigned Figure 2 to ensure all subfigures are clear and readable. 

Comment 2. It's confusing that Z has been represented as a vector in Algorithm 1 (but it's actually a 2D matrix)? The symbols and notations used in the algorithm should be defined clearly.

Response: 

We have revised Algorithm 1 and provided a clear definition for matrix Z (see Algorithm 1 and lines 280-281 on page 8). We have also included a list of symbols in Table 1 on Page 7.

Comment 3. The authors claim, in section 5.3, that the proposed method outperforms both STOMP and Series2Graph methods. This doesn't really give an accurate picture of the comparison. Out of 5 datasets, the proposed method outperforms Series2Graph on only one and performs the same on the remaining datasets. The authors should revise their claim and make it more specific. The one dataset on which the proposed method performs better, it does 20% better than Series2Graph which is satisfactory. However, if we consider the overall performance comparison shown in Figure 10, the proposed method does not do significantly better than Series2Graph. It would be more appealing if the authors used more datasets and could show that their proposed method performs better on at least about 50% of the datasets used in the experiment. However, considering the scalability factor, the proposed method does have some useful contributions.

Response: 

To enhance the evaluation, we have added more datasets (100 time series for seven groups of UCR datasets listed in Table 4) to compare our method with the Series2Graph method. Now the experimental results show that the proposed GrapTS outperforms Series2Graph in total accuracy while spending less runtime; that is, GrapTS absolutely excels Series2Graph in seven groups and achieves the same accuracy in two groups. (see Section 5.5, Figure 13 and S1 Table). We also have revised our claim in Section 5.3 (see lines 489-492 on page 15).

Comment 4. Minor issues: Some grammar and spelling errors exist. For example, it should be "rest", not "reset", in the paragraph under Table 1.

Response: 

We have corrected the errors as suggested. Moreover, we checked the whole paper for grammar, spelling and punctuation mistakes and made the corrections in the revised version.

---

## [Decision Letter · Decision Letter 1]

7 Jun 2023

PONE-D-22-15666R1GraphTS: Graph-Represented Time Series for Subsequence Anomaly DetectionPLOS ONE

Dear Dr. Huang,

Thank you for submitting your manuscript to PLOS ONE. After careful consideration, we feel that it has merit but does not fully meet PLOS ONE’s publication criteria as it currently stands. Therefore, we invite you to submit a revised version of the manuscript that addresses the points raised during the review process.

We look forward to receiving your revised manuscript.

Kind regards,

Vijayalakshmi Kakulapati, Ph.D

Academic Editor

PLOS ONE

Journal Requirements:

Reviewers' comments:

Reviewer's Responses to Questions

**Comments to the Author**

1. If the authors have adequately addressed your comments raised in a previous round of review and you feel that this manuscript is now acceptable for publication, you may indicate that here to bypass the “Comments to the Author” section, enter your conflict of interest statement in the “Confidential to Editor” section, and submit your "Accept" recommendation.

Reviewer #2: All comments have been addressed

Reviewer #3: (No Response)

Reviewer #4: All comments have been addressed

2. Is the manuscript technically sound, and do the data support the conclusions?

Reviewer #2: Yes

Reviewer #3: (No Response)

Reviewer #4: Yes

3. Has the statistical analysis been performed appropriately and rigorously? 

Reviewer #2: N/A

Reviewer #3: (No Response)

Reviewer #4: Yes

4. Have the authors made all data underlying the findings in their manuscript fully available?

Reviewer #2: Yes

Reviewer #3: (No Response)

Reviewer #4: (No Response)

5. Is the manuscript presented in an intelligible fashion and written in standard English?

Reviewer #2: Yes

Reviewer #3: (No Response)

Reviewer #4: (No Response)

6. Review Comments to the Author

Reviewer #2: This article proposes a graph-based anomaly detection method for time series data. The proposed method, GraphTS, aims to do efficient detection of both recurrent and rare anomalies.

I'll repeat from my first review that this article is well-organized, well-explained, and easy to read.

I had several major concerns that the authors have addressed in their revised manuscript. One of which was about the lack of enough data and some claims they made that I didn't find convincing.

However, I'm satisfied with their explanation and revision.

Reviewer #3: Author should add keyword list.keywors list contain 5 to 8 words.

Conclusion to be made more systematic and future scope to be elaborated more on technical features that are planned to be added in the proposed system in the near future.

author add more referecne in introducation as below

1-Anomaly Detection via Correlation Clustering

2-Attack-Specific Feature Selection for Anomaly Detection in Software-Defined Networks

3-Two-phase classification: ANN and A-SVM classifiers on motor imagery BCI

Authors should further explain equations and maths. It is too hard to udnerstnad at the moemnt. Secondly, if these are general maths easily available on the internet, then authors should remove it and add reference instread.

The use of English language is fine, however, it is recommended to be checked once again.

Reviewer #4: (No Response)

7. PLOS authors have the option to publish the peer review history of their article (what does this mean?). If published, this will include your full peer review and any attached files.

Reviewer #2: No

Reviewer #3: No

Reviewer #4: No

---

## [Author Response · Author response to Decision Letter 1]

18 Jul 2023

Response to reviewers’ comments for: GraphTS: Graph-Represented Time Series for Subsequence Anomaly Detection

We thank the editor and all the reviewers for their constructive feedback to help improve this paper. Below please find the point-to-point responses to the comments raised by the reviewers. Accordingly, we highlighted our modifications in the new version.

Response to Reviewer 2

Comment 1: This article proposes a graph-based anomaly detection method for time series data. The proposed method, GraphTS, aims to do efficient detection of both recurrent and rare anomalies. I'll repeat from my first review that this article is well-organized, well-explained, and easy to read. I had several major concerns that the authors have addressed in their revised manuscript. One of which was about the lack of enough data and some claims they made that I didn't find convincing. However, I'm satisfied with their explanation and revision.

Response: 

Thank you for your positive feedback on our revised manuscript. We appreciate your contribution to improving our work.

Response to Reviewer 3

Comment 1. Author should add keyword list. keywords list contain 5 to 8 words.

Response: 

As suggested, we have added a keywords list in the revised paper (see keywords section, page 1). 

Comment 2. Conclusion to be made more systematic and future scope to be elaborated more on technical features that are planned to be added in the proposed system in the near future.

Response: 

We have revised the conclusion section (see conclusion section, page 17) and elaborated on the future work in the revised paper (see lines 559-572 on page 17).

Comment 3. author add more reference in introduction as below

1-Anomaly Detection via Correlation Clustering

2-Attack-Specific Feature Selection for Anomaly Detection in Software-Defined Networks

3-Two-phase classification: ANN and A-SVM classifiers on motor imagery BCI

Response: 

As suggested, we have added more related works (i.e., [45] [46] [47]) to the introduction section. 

Comment 4. Authors should further explain equations and maths. It is too hard to understand at the moment. Secondly, if these are general maths easily available on the internet, then authors should remove it and add reference instead.

Response: 

We have added explanation for Eq. (1) and cited reference for Definitions 1-4.

Comment 5. The use of English language is fine, however, it is recommended to be checked once again.

Response: 

We have checked the whole paper for grammar, spelling and punctuation mistakes and made the corrections in the revised version.

---

## [Decision Letter · Decision Letter 2]

2 Aug 2023

GraphTS: Graph-Represented Time Series for Subsequence Anomaly Detection

PONE-D-22-15666R2

Dear Dr. Huang,

We’re pleased to inform you that your manuscript has been judged scientifically suitable for publication and will be formally accepted for publication once it meets all outstanding technical requirements.

Kind regards,

Vijayalakshmi Kakulapati, Ph.D

Academic Editor

PLOS ONE

Reviewers' comments:

Accept

---

## [Editor Report · Acceptance letter]

7 Aug 2023

PONE-D-22-15666R2 

GraphTS: Graph-Represented Time Series for Subsequence Anomaly Detection 

Dear Dr. Huang:

I'm pleased to inform you that your manuscript has been deemed suitable for publication in PLOS ONE. Congratulations! Your manuscript is now with our production department. 

Kind regards, 

on behalf of

Dr. Vijayalakshmi Kakulapati 

Academic Editor

PLOS ONE